# Bisimulation Metrics are Optimal Transport Distances, and Can be Computed Efficiently

**Sergio Calo   Anders Jonsson   Gergely Neu   Ludovic Schwartz   Javier Segovia-Aguas**

Universitat Pompeu Fabra, Barcelona, Spain

`{sergio.calo,anders.jonsson,gergely.neu,ludovic.schwartz,javier.segovia}@upf.edu`

## Abstract

We propose a new framework for formulating optimal transport distances between Markov chains. Previously known formulations studied couplings between the entire joint distribution induced by the chains, and derived solutions via a reduction to dynamic programming (DP) in an appropriately defined Markov decision process. This formulation has, however, not led to particularly efficient algorithms so far, since computing the associated DP operators requires fully solving a static optimal transport problem, and these operators need to be applied numerous times during the overall optimization process. In this work, we develop an alternative perspective by considering couplings between a "flattened" version of the joint distributions that we call discounted occupancy couplings, and show that calculating optimal transport distances in the full space of joint distributions can be equivalently formulated as solving a linear program (LP) in this reduced space. This LP formulation allows us to port several algorithmic ideas from other areas of optimal transport theory. In particular, our formulation makes it possible to introduce an appropriate notion of entropy regularization into the optimization problem, which in turn enables us to directly calculate optimal transport distances via a Sinkhorn-like method we call Sinkhorn Value Iteration (SVI). We show both theoretically and empirically that this method converges quickly to an optimal coupling, essentially at the same computational cost of running vanilla Sinkhorn in each pair of states. Along the way, we point out that our optimal transport distance exactly matches the common notion of bisimulation metrics between Markov chains, and thus our results also apply to computing such metrics, and in fact our algorithm turns out to be significantly more efficient than the best known methods developed so far for this purpose.

## 1   Introduction

Measuring distances between structured objects and sequences is an important problem in a variety of areas of science. The more structured the objects become, the harder it gets to define appropriate notions of distances, as good notions of proximity need to take into account the possibly complex relationships between the constituent parts of each object. The possibility that the objects in question may be random further complicates the picture, and in such cases it becomes more natural to measure distances between the underlying joint probability distributions. Within the specific context of comparing stochastic processes, two natural notions of distance have emerged over the past decades: the notion of *probabilistic bisimulation metrics* that takes its root in modal logic and theoretical computer science [Sangiorgi, 2009, Abate, 2013], and the notion of *optimal-transport distances* that originates from probability theory [Villani, 2009, Peyré and Cuturi, 2019]. In this paper, we show that bisimulation metrics are in fact optimal-transport distances, and we make use of this observation to derive efficient algorithms for computing distances between stochastic processes.

38th Conference on Neural Information Processing Systems (NeurIPS 2024).

The two distance notions have found strikingly different applications. Bisimulation emerged within the area of theoretical computer science as one of the most important important concepts in concurrency theory and formal verification of computer systems [Park, 1981, Milner, 1989], and has been extended to probabilistic transition systems by Larsen and Skou [1989]. Within machine learning, bisimulation metrics have become especially popular in the context of reinforcement learning (RL) due to the work of Ferns et al. [2004], and have become one of the few standard tools of representation learning [Jiang, 2018, 2024]. In particular, the work of Ferns et al. [2004] advocates for using bisimilarity as a basis for state aggregation, measuring similarities of states in terms of similarities of two chains $M_{\mathcal{X}}$ and $M_{\mathcal{Y}}$ that only differ in their initial state. While this approach has inspired numerous follow-up works [Gelada et al., 2019, Castro, 2020, Agarwal et al., 2021b, Zhang et al., 2021, Hansen-Estruch et al., 2022, Castro et al., 2022], ultimately this line of work has failed to discover efficient algorithms for computing bisimulation metrics and has largely resorted to heuristics for computing similarity metrics.

On the other hand, optimal transport (OT) has found numerous applications in areas as diverse as economics [Galichon, 2016], signal processing [Kolouri et al., 2017], or genomics [Schiebinger et al., 2019]. Within machine learning, it has been used for the similarly diverse areas of domain adaptation [Courty et al., 2016], generative modeling [Arjovsky et al., 2017, Song et al., 2020, Shi et al., 2024], representation learning [Courty et al., 2018], and, perhaps most relevant to our work, as a way of measuring distances between graphs [Titouan et al., 2019, Chen et al., 2022, Chuang and Jegelka, 2022]. The recent works of Yi et al. [2021], Brugère et al. [2024] propose to define graph distances via studying the behavior of random walks defined on the graph, thus reducing the problem of comparing graphs to comparing stochastic processes—exactly the subject of the present paper. Other applications of OT between stochastic processes include generative modeling for sequential data [Xu et al., 2020], pricing and hedging in mathematical finance [Backhoff-Veraguas et al., 2017], and analyzing multistage stochastic optimization problems [Pflug, 2010, Bartl and Wiesel, 2022]. It appears however that the literature on optimal transport for stochastic processes has apparently not yet discovered connections with bisimulation metrics and the rich intellectual history behind it. Also, applications of optimal transport for representation learning within the context of reinforcement learning appear to be nonexistent.

In this paper we observe that, despite their apparent differences, bisimulation metrics and optimal transport distances are one and the same. Furthermore, we provide a new perspective on both OT distances and bisimulation metrics by formulating the distance metric as the solution of a linear program (LP) in the space of "occupancy couplings", a finite-dimensional projection of the infinite-dimensional process laws. Building on tools from computational optimal transport [Peyré and Cuturi, 2019] and entropy-regularized Markov decision processes [Neu et al., 2017, Geist et al., 2019], we design an algorithm that effectively combines Sinkhorn's algorithm [Sinkhorn and Knopp, 1967, Cuturi, 2013] with an entropy-regularized version of the classic Value Iteration algorithm [Bellman, 1957, Neu et al., 2017]. Building on recent work on computational optimal transport [Altschuler et al., 2017, Ballu and Berthet, 2023], we provide theoretical guarantees for the resulting algorithm (called Sinkhorn Value Iteration) and perform numerical studies that demonstrate its effectiveness for computing distances between Markov chains.

**Notations.** For a finite set $\mathcal{S}$, we use $\Delta_{\mathcal{S}}$ to denote the set of all probability distributions over $\mathcal{S}$. We will denote infinite sequences by $\overline{x} = (x_0, x_1, \dots)$ and the corresponding subsequences as $\overline{x}_n = (x_0, x_1, \dots, x_n)$. For two sets $\mathcal{X}$ and $\mathcal{Y}$, we will often write $\mathcal{X}\mathcal{Y}$ to abbreviate the direct-product notation $\mathcal{X} \times \mathcal{Y}$, and for two indices $x$ and $y$ and a function $f : \mathcal{X}\mathcal{Y} \to \mathcal{Z}$, we will often write $f(xy)$ instead of $f(x,y)$ to save space. Also, we will denote scalar products by $\langle \cdot, \cdot \rangle$ and use $\|\cdot\|_p$ to denote the $\ell_p$-norm.

## 2 Preliminaries

We study the problem of measuring distances between pairs of finite Markov chains. Specifically, we consider two stationary Markov processes $M_{\mathcal{X}} = (\mathcal{X}, P_{\mathcal{X}}, \nu_{0,\mathcal{X}})$ and $M_{\mathcal{Y}} = (\mathcal{Y}, P_{\mathcal{Y}}, \nu_{0,\mathcal{Y}})$, where

- $\mathcal{X}$ and $\mathcal{Y}$ are the finite state spaces with cardinalities $m = |\mathcal{X}|$ and $n = |Y|$,
- $P_{\mathcal{X}} : \mathcal{X} \to \Delta_{\mathcal{X}}$ and $P_{\mathcal{Y}} : \mathcal{Y} \to \Delta_{\mathcal{Y}}$ are the transition kernels that determine the evolution of the states as $P_{\mathcal{X}}(x'|x) = \mathbb{P}\left[X_{t+1} = x' \mid X_t = x\right]$ and $P_{\mathcal{Y}}(y'|y) = \mathbb{P}\left[Y_{t+1} = y' \mid Y_t = y\right]$ for all $t$, and
- $\nu_{0,\mathcal{X}}$ and $\nu_{0,\mathcal{Y}}$ are the initial-state distributions with $X_0 \sim \nu_{0,\mathcal{X}}$ and $Y_0 \sim \nu_{0,\mathcal{Y}}$.

Without significant loss of generality, we will suppose that the initial states are fixed almost surely as $X_0 = x_0$ and $Y_0 = y_0$, and refer to their corresponding joint distribution as $\nu_0 = \delta_{x_0, y_0}$. These objects together define a sequence of joint distributions $\mathbb{P}\left[(X_0, X_1, \ldots, X_n) = (x_0, x_1, \ldots, x_n)\right]$ and $\mathbb{P}\left[(Y_0, Y_1, \ldots, Y_n) = (y_0, y_1, \ldots, y_n)\right]$ for each $n$, which together define respective the laws of the infinite sequences $\overline{X} = (X_1, X_2, \ldots)$ and $\overline{Y} = (Y_1, Y_2, \ldots)$ via Kolmogorov's extension theorem. With a slight abuse of notation, we will use $M_{\mathcal{X}}$ and $M_{\mathcal{Y}}$ to denote the corresponding measures that satisfy $M_{\mathcal{X}}(\overline{x}_n) = \mathbb{P}\left[\overline{X}_n = \overline{x}_n\right]$ and $M_{\mathcal{Y}}(\overline{y}_n) = \mathbb{P}\left[\overline{Y}_n = \overline{y}_n\right]$ for any $\overline{x} \in \mathcal{X}^\infty$, $\overline{y} \in \mathcal{Y}^\infty$ and $n$. The corresponding conditional distributions are denoted as $M_{\mathcal{X}}(x_n|\overline{x}_{n-1}) = \mathbb{P}\left[X_n = x_n|\overline{X}_{n-1} = \overline{x}_{n-1}\right]$ and $M_{\mathcal{Y}}(y_n|\overline{y}_{n-1}) = \mathbb{P}\left[Y_n = y_n|\overline{Y}_{n-1} = \overline{y}_{n-1}\right]$.

## 2.1 Optimal transport between Markov chains

Our main object of interest in this work is a notion of optimal transport distance between infinite-horizon Markov chains. Several previous works have studied such distances (which are discussed in detail in Appendix A), and our precise definition we give below is closest to Moulos [2021], O'Connor et al. [2022] and Brugère et al. [2024]. We consider Markov chains on state spaces where a "ground metric" (or "ground cost") $c : \mathcal{X} \times \mathcal{Y} \to \mathbb{R}^+$ is available to measure distances between any two individual states $x \in \mathcal{X}$ and $y \in \mathcal{Y}$, with the distance denoted as $c(x, y)$. For any two sequences $\overline{x} = (x_1, x_2, \ldots)$ and $\overline{y} = (y_1, y_2, \ldots)$, we define the discounted total cost

$$c_\gamma(\overline{x}, \overline{y}) = \sum_{t=0}^{\infty} \gamma^t c(x_t, y_t), \tag{1}$$

where $\gamma \in (0, 1)$ is the *discount factor* that expresses the preference that two sequences be considered further apart if they exhibit differences at earlier times in terms of the ground cost $c$. Following the optimal-transport literature, we will consider distances between the stochastic processes $M_{\mathcal{X}}$ and $M_{\mathcal{Y}}$ via the notion of *couplings*. To this end, we define a coupling of $M_{\mathcal{X}}$ and $M_{\mathcal{Y}}$ as a stochastic process evolving on the joint space $\mathcal{XY}$, with its law defined for all $n$ as $M_{\mathcal{XY}}(\overline{x}_n\overline{y}_n) = \mathbb{P}\left[\overline{X}_n = \overline{x}_n, \overline{Y}_n = \overline{y}_n\right]$, required to satisfy $\sum_{\overline{y}_n \in \mathcal{Y}^n} M_{\mathcal{XY}}(\overline{x}_n\overline{y}_n) = M_{\mathcal{X}}(\overline{x}_n)$ and $\sum_{\overline{x}_n \in \mathcal{X}^n} M_{\mathcal{XY}}(\overline{x}_n\overline{y}_n) = M_{\mathcal{Y}}(\overline{y}_n)$. We will define the set of all such couplings as $\Pi$.

The notion of couplings defined above does not respect the temporal structure of the Markov chains $M_{\mathcal{X}}$ and $M_{\mathcal{Y}}$ appropriately: while by definition the distribution of state $X_n$ may only be causally influenced by past states $X_k$ with $k < n$, the general notion of coupling above allows the state $X_n$ to be influenced by future states $Y_k$ with $k \geq n$ as well. To rule out this possibility (and following past works mentioned in the introduction), we will introduce the notion of *bicausal couplings*. A coupling $M_{\mathcal{XY}}$ is called bicausal if and only if it satisfies

$$\sum_{y_n} M_{\mathcal{XY}}(x_ny_n|\overline{x}_{n-1}\overline{y}_{n-1}) = M_{\mathcal{X}}(x_n|\overline{x}_{n-1}) \quad \text{and} \quad \sum_{x_n} M_{\mathcal{XY}}(x_ny_n|\overline{x}_{n-1}\overline{y}_{n-1}) = M_{\mathcal{Y}}(y_n|\overline{y}_{n-1})$$

for all sequences $\overline{x}, \overline{y} \in \mathcal{X}^\infty \times \mathcal{Y}^\infty$ and all $n$. Denoting the set of all bicausal couplings by $\Pi_{\text{bc}}$, we define our optimal transport distance as

$$\mathbb{W}_\gamma(M_{\mathcal{X}}, M_{\mathcal{Y}}; c, x_0, y_0) = \inf_{M_{\mathcal{XY}} \in \Pi_{\text{bc}}} \int c_\gamma(\overline{X}, \overline{Y}) \, dM_{\mathcal{XY}}(\overline{X}, \overline{Y}), \tag{2}$$

where we emphasize the dependence of the distance on $x_0, y_0$ explicitly with our notation.

By noticing that the optimization problem outlined above can be reformulated as a Markov decision process (MDP), Moulos [2021] has shown that the infimum in (2) is achieved within the family of Markovian couplings that satisfy

$$\sum_{y_n} M_{\mathcal{XY}}(x_ny_n|\overline{x}_{n-1}\overline{y}_{n-1}) = P_{\mathcal{X}}(x_n|x_{n-1}) \quad \text{and} \quad \sum_{x_n} M_{\mathcal{XY}}(x_ny_n|\overline{x}_{n-1}\overline{y}_{n-1}) = P_{\mathcal{Y}}(y_n|y_{n-1})$$

for all sequences of state pairs and all values of $n$. Furthermore, it can be seen that Markovian couplings are fully specified in terms of *transition couplings* of the form $\pi : \mathcal{XY} \to \Delta_{\mathcal{XY}}$, with $\pi(x'y'|xy)$ standing for $\mathbb{P}\left[(X_{t+1}, Y_{t+1}) = (x', y')|(X_t, Y_t) = (x_t, y_t)\right]$ under the law induced by the coupling. We say that a transition coupling is *valid* if it satisfies the marginal constraints $\sum_{y'} \pi(x'y'|xy) = P_{\mathcal{X}}(x'|x)$ and $\sum_{x'} \pi(x'y'|xy) = P_{\mathcal{Y}}(y'|y)$. Defining the set of such valid transition couplings in state pair $xy$ by $\Pi_{xy} = \left\{p \in \Delta_{\mathcal{XY}} : \sum_{y'} p(x'y') = P_{\mathcal{X}}(x'|x), \sum_{x'} p(x'y') = \right.$

$P_{\mathcal{Y}}(y'|y)\}$, Moulos [2021] introduces an MDP $\mathcal{M}$ with an infinite action set corresponding to picking the joint next-state couplings in $\Pi_{xy}$. An optimal transition coupling can be found by solving the following Bellman optimality equations of the MDP $\mathcal{M}$:

$$V^*(xy) = c(xy) + \gamma \inf_{p \in \Pi_{xy}} \sum_{x'y'} p(x'y')V^*(x'y'). \tag{3}$$

The infimum on the right-hand side is achieved by an optimal transition coupling $\pi^*(\cdot|xy) = \arg\min_{p \in \Pi_{xy}} \sum_{x'y'} p(x'y')V^*(x'y')$. The solution $V^*$ is unique and can be shown to satisfy $V^*(xy) = \mathbb{W}_\gamma(M_{\mathcal{X}}, M_{\mathcal{Y}}; c, x, y)$ for all $xy \in \mathcal{X}\mathcal{Y}$. For completeness, we include the precise definition of the MDP $\mathcal{M}$ and the proofs of these results in Appendix B.

## 2.2 Bisimulation metrics

The notion of *bisimulation metrics* has been introduced by Desharnais et al. [1999, 2004] and van Breugel and Worrell [2001], with the purpose of defining distances between Markov chains, using a methodology rooted in modal logic that at first may appear entirely different from the optimal-transport framework described above. We only give a very high-level overview of the classic logic-based characterization here (as the fine details are irrelevant to the final conclusion that this section is headed to), and refer the reader to the additional discussion in Appendix A for further reading. Desharnais et al. [1999, 2004] considered *labeled* Markov chains where a labeling function $r : \mathcal{X} \cup \mathcal{Y} \to \mathbb{R}$ assigns labels to each state, and defined bisimulation metrics via

$$d_\gamma(M_{\mathcal{X}}, M_{\mathcal{Y}}; r, x_0, y_0) = \sup_{f \in \mathcal{F}_\gamma} |f_{M_{\mathcal{X}}}(x_0) - f_{M_{\mathcal{Y}}}(y_0)|, \tag{4}$$

where $\mathcal{F}_\gamma$ is a family of functional expressions generated by a certain grammar, and $f_{M_{\mathcal{X}}} : \mathcal{X} \to \mathbb{R}$ and $f_{M_{\mathcal{Y}}} : \mathcal{Y} \to \mathbb{R}$ are the respective "interpretations" of each $f \in \mathcal{F}$ on the Markov chains $M_{\mathcal{X}}$ and $M_{\mathcal{Y}}$. Building on this formulation, van Breugel and Worrell [2001] have shown that the distance metric can be equivalently characterized by the solution of a fixed-point equation, whose expression was subsequently used by Ferns et al. [2004] to define bisimulation metrics for Markov *decision* processes where $r$ takes the role of a reward function, and the evolution of states may be influenced by actions. The case of having no actions available corresponds to our setting, where their definition of a bisimulation metric simplifies to the solution of the fixed-point equation

$$U^*(xy) = (1 - \gamma)|r(x) - r(y)| + \gamma \inf_{p \in \Pi_{xy}} \sum_{x'y'} p(x'y')U^*(x'y'). \tag{5}$$

The solution to this system is unique and satisfies $U^*(xy) = d_\gamma(M_{\mathcal{X}}, M_{\mathcal{Y}}; r, x, y)$. Putting this result side-by-side with Equation (3), one can immediately realize that *bisimulation metrics and our notion of optimal transport distances coincide when picking the ground cost function $c(xy) = (1 - \gamma)|r(x) - r(y)|$*. To our knowledge, this remarkable observation has not been publicly made anywhere in either the optimal-transport or the bisimulation-metric literature. This connection has several important implications, which we have already discussed at some length in Section 1. We relegate further discussion of these metrics in the light of this observation to Appendix A.

## 3 Optimal transport between Markov chains as a linear program

Optimal transport problems can typically be formulated as linear programs (LPs), since couplings can be characterized as joint distributions satisfying a set of easily-expressed linear constraints (see, e.g., Chapter 3 in Peyré and Cuturi 2019). Our problem is no exception, and in fact the original problem statement of Equation 2 can be expressed in this form: the constraints defining $\Pi_{\text{bc}}$ are all linear in $M_{\mathcal{X}\mathcal{Y}}$. However, $M_{\mathcal{X}\mathcal{Y}}$ is an infinite-dimensional object and thus this formulation is not instructive for developing computationally tractable algorithms. We address this problem in this section, where we define an equivalent LP formulation that replaces the infinite-dimensional optimization variable with an appropriate low-dimensional projection. Our framework builds on the classic LP formulation of optimal control in MDPs first proposed in the 1960's [Manne, 1960, de Ghellinck, 1960, d'Epenoux, 1963, Denardo, 1970], and covered thoroughly in several standard textbooks (e.g., Section 6.9 of Puterman 1994).

In order to set things up, we need to start with some important definitions. We say that a transition coupling $\pi : \mathcal{X}\mathcal{Y} \to \Delta_{\mathcal{X}\mathcal{Y}}$ generates a trajectory $(X_0, Y_0, X_1, Y_1, \dots)$ if $(X_0, Y_0) \sim \nu_0$ and the

subsequent state-pairs are drawn independently from the transition coupling as $(X_{t+1}, Y_{t+1}) \sim \pi(\cdot|X_t, Y_t)$ for all $t$. Then, we define the *occupancy coupling* $\mu^\pi$ associated with this process as a distribution over $\mathcal{XY} \times \mathcal{XY}$, with each of its entries $xy, x'y'$ defined as

$$\mu^\pi(xy, x'y') = (1 - \gamma)\mathbb{E}_\pi\left[\sum_{t=0}^\infty \gamma^t \mathbb{I}_{\{(X_t, Y_t) = (x,y), (X_{t+1}, Y_{t+1}) = (x', y')\}}\right],$$

where $\mathbb{E}_\pi[\cdot]$ emphasizes that the trajectory of state-pairs has been generated by $\pi$. In words, $\mu^\pi(xy, x'y')$ is the discounted number of times that the quadruple $(xy, x'y')$ is visited by the process. With this definition, it is easy to notice that the objective optimized in Equation (2) can be rewritten as a linear function of $\mu^\pi$. Indeed, suppose that $M_{\mathcal{XY}}$ is the law of the process generated by $\pi$ as described above, so that we can write

$$\int c_\gamma(\overline{X}, \overline{Y})\, \mathrm{d}M_{\mathcal{XY}}(\overline{X}, \overline{Y}) = \mathbb{E}_\pi\left[\sum_{t=0}^\infty \gamma^t c(X_t, Y_t)\right] = \mathbb{E}_\pi\left[\sum_{t=0}^\infty \gamma^t \sum_{xy} \mathbb{I}_{\{(X_t, Y_t) = (x,y)\}} c(xy)\right]$$

$$= \sum_{xy} \mathbb{E}_\pi\left[\sum_{t=0}^\infty \gamma^t \mathbb{I}_{\{(X_t, Y_t) = (x,y)\}}\right] c(xy) = \frac{1}{1 - \gamma} \sum_{xy, x'y'} \mu^\pi(xy, x'y') c(xy) = \frac{\langle \mu^\pi, c \rangle}{1 - \gamma}.$$

We say that an occupancy coupling $\mu$ is *valid* if it is generated by a valid transition coupling. It is easy to verify that every valid occupancy coupling $\mu \in \Delta_{\mathcal{XY} \times \mathcal{XY}}$ satisfies the following three constraints:

$$\sum_{x'y'} \mu(xy, x'y') = \gamma \sum_{x'y'} \mu(x'y', xy) + (1 - \gamma)\nu_0(xy) \qquad (\forall xy \in \mathcal{XY}), \tag{6}$$

$$\sum_{y'} \mu(xy, x'y') = \sum_{x''y''} \mu(xy, x''y'') P_\mathcal{X}(x'|x) \qquad (\forall x, x' \in \mathcal{X} \times \mathcal{X}), \tag{7}$$

$$\sum_{x'} \mu(xy, x'y') = \sum_{x''y''} \mu(xy, x''y'') P_\mathcal{Y}(y'|y) \qquad (\forall y, y' \in \mathcal{Y} \times \mathcal{Y}). \tag{8}$$

We refer to the first equality constraint as the *flow constraint* and the second and third ones as the *transition coherence constraints* for $M_\mathcal{X}$ and $M_\mathcal{Y}$, respectively[1]. We show in the following lemma that the above conditions uniquely characterize the set of valid occupancy couplings.

**Lemma 1.** *A distribution $\mu \in \Delta_{\mathcal{XY} \times \mathcal{XY}}$ is a valid occupancy coupling associated with some transition coupling $\pi : \mathcal{XY} \to \Delta_{\mathcal{XY}}$ if and only if it satisfies Equations (6)–(8).*

One important consequence of this result is that for each transition coupling, one can associate a transition coupling $\pi_\mu$ with $\mu^{\pi_\mu} = \mu$, with entries satisfying $\sum_{x''y''} \mu(xy, x''y'')\pi_\mu(x'y'|xy) = \mu(xy, x'y')$. A proof is provided in Appendix B.2. Having established in the previous section that stationary occupancy couplings are sufficient to achieve the supremum in the definition of the OT distance of Equation (2), this result immediately implies the following.

**Theorem 1.** *Let $\mathcal{B}$, $\mathcal{S}_\mathcal{X}$ and $\mathcal{S}_\mathcal{Y}$ respectively stand for the sets of all distributions $\mu \in \Delta_{\mathcal{XY} \times \mathcal{XY}}$ that satisfy equations (6), (7), and (8). Then,*

$$\mathbb{W}_\gamma(M_\mathcal{X}, M_\mathcal{Y}; c, x_0, y_0) = \frac{1}{1 - \gamma} \cdot \inf_{\mu \in \mathcal{B} \cap \mathcal{S}_\mathcal{X} \cap \mathcal{S}_\mathcal{Y}} \langle \mu, c \rangle.$$

Since the constraints on $\mu$ in the above reformulation are all linear, the optimization problem stated above is obviously a linear program. Notably, the resulting LP is *not* the standard dual to the LP associated with the MDP formulation introduced in Section 2.1, which would result in an infinite-dimensional LP with one constraint associated with each of the continuously-valued actions. Such an infinite-dimensional reformulation was previously considered by Chen et al. [2012] in the context of computing bisimulation metrics, who used it as a tool for analysis rather than algorithm design. Notably, our formulation results in a finite-dimensional LP with $|\mathcal{X}|^2 |\mathcal{Y}|^2$ variables and $|\mathcal{X}|^2 + |\mathcal{X}||\mathcal{Y}| + |\mathcal{Y}|^2$ constraints. Building on the celebrated result of Ye [2011] (and similarly to the work of Chen et al., 2012 mentioned above), one can show that our LP can be solved in strongly polynomial time via an appropriate adaptation of the simplex method or the classic policy iteration method of Howard [1960]. We propose an alternative methodology in the next section.

---

[1] When $\gamma = 1$, the first condition is known as *detailed balance* within statistical physics. Altogether, the constraints closely resemble what are often called the "Bellman flow constraints" in today's RL literature.

# 4 Sinkhorn Value Iteration

Solving optimal-transport problems via standard LP solvers (like variations of the simplex method or interior-point methods) is known to be empirically hard, and thus we seek alternatives to this approach towards optimizing our own LP defined in the previous section. In computational optimal transport, a paradigm shift was initiated by Cuturi [2013] who successfully applied entropic regularization to the classic LP objective of optimal transport, and solved the resulting optimization method through an iterative algorithm called the Sinkhorn–Knopp method (sometimes simply called Sinkhorn's algorithm, due to Sinkhorn and Knopp, 1967). This method is based on finding a feasible point in a two-constraint problem by alternately satisfying one and the other, and has resulted in practical algorithms that were orders of magnitude faster than all previously studied methods. Entropy regularization and Sinkhorn's algorithm have thus became the most important cornerstones of computational optimal transport. Drawing on the same principles as well as the theory of entropy-regularized Markov decision processes [Neu et al., 2017, Geist et al., 2019], we develop a computationally effective algorithm for computing optimal transport distances between Markov chains below.

## 4.1 Formal definition: Mirror Sinkhorn on the space of occupancy couplings

Our method is an adaptation of a version of Sinkhorn's algorithm called Mirror Sinkhorn, first proposed and analyzed by Ballu and Berthet [2023]. This method combines mirror-descent-style updates [Nemirovski and Yudin, 1983, Beck and Teboulle, 2003] with alternating projections to two convex sets whose intersection corresponds to the feasible set we seek to optimize over. In our adaptation, we choose the two sets as

$$\mathcal{B}_{\mathcal{X}} = \left\{ \mu : \sum_{y'} \mu(xy, x'y') = \left( \gamma \sum_{x''y''} \mu(x''y'', xy) + (1-\gamma)\nu_0(xy) \right) P_{\mathcal{X}}(x'|x) \quad (\forall xy, x') \right\},$$

that can be seen to be the set of distributions $\mu$ that satisfy both Equations (6) and (7), and

$$\mathcal{B}_{\mathcal{Y}} = \left\{ \mu : \sum_{x'} \mu(xy, x'y') = \left( \gamma \sum_{x''y''} \mu(x''y'', xy) + (1-\gamma)\nu_0(xy) \right) P_{\mathcal{Y}}(y'|y) \quad (\forall xy, y') \right\}$$

which is the set of distributions $\mu$ that satisfy both Equations (6) and (8). Naturally, the intersection of the two sets corresponds to valid occupancy couplings. It remains to define an appropriate notion of entropy for the purpose of regularization. Following Neu et al. [2017], we will use the *conditional relative entropy* defined between two joint distributions $\mu, \mu' \in \Delta_{\mathcal{XYXY}}$ as

$$\mathcal{H}(\mu\|\mu') = \sum_{xy,x'y'} \mu(xy, x'y') \log \frac{\mu(xy, x'y') / \sum_{x''y''} \mu(xy, x''y'')}{\mu'(xy, x'y') / \sum_{x''y''} \mu'(xy, x''y'')}$$

$$= \sum_{xy,x'y'} \mu(xy, x'y') \log \frac{\pi_\mu(x'y'|xy)}{\pi_{\mu'}(x'y'|xy)}.$$

It is an easy exercise to show that $\mathcal{H}$ is a Bregman divergence that is convex in its first argument (see, e.g., Appendix A.1 of Neu et al., 2017). Note however that $\mathcal{H}(\mu\|\mu')$ can be zero even if $\mu \neq \mu'$ and thus it is not strongly convex in $\mu$.

With these ingredients, we symbolically define our algorithm as calculating the sequence of updates

$$\mu_{k+1} = \underset{\mu \in \mathcal{B}_k}{\arg\min} \left\{ \langle \mu, c \rangle + \frac{1}{\eta} \mathcal{H}(\mu\|\mu_k) \right\} \tag{9}$$

for each $k = 1, \ldots, K-1$, where $\mu_1$ is the occupancy coupling associated to the trivial coupling $\pi_1(\cdot|xy) = P_{\mathcal{X}}(\cdot|x) \otimes P_{\mathcal{Y}}(\cdot|y)$ for each state pair $xy \in \mathcal{XY}$, $\eta > 0$ is a stepsize (or learning-rate) parameter and $\mathcal{B}_k$ is chosen to be $\mathcal{B}_{\mathcal{X}}$ in odd rounds and $\mathcal{B}_{\mathcal{Y}}$ in even rounds. By adapting tools from the theory of entropy-regularized Markov decision processes, the updates can be computed in closed form by solving a system of equations closely resembling the regularized Bellman equations. In particular, we define the *Bellman–Sinkhorn operators* for a given transition coupling $\pi$ as the operators

$\mathcal{T}_{\mathcal{X}}^{\pi} : \mathbb{R}^{\mathcal{X}\mathcal{Y}\times\mathcal{X}} \to \mathbb{R}^{\mathcal{X}\mathcal{Y}\times\mathcal{X}}$ and $\mathcal{T}_{\mathcal{Y}}^{\pi} : \mathbb{R}^{\mathcal{X}\mathcal{Y}\times\mathcal{Y}} \to \mathbb{R}^{\mathcal{X}\mathcal{Y}\times\mathcal{Y}}$ acting on functions $V_{\mathcal{X}} \in \mathbb{R}^{\mathcal{X}\mathcal{Y}\times\mathcal{X}}$ and $V_{\mathcal{Y}} \in \mathbb{R}^{\mathcal{X}\mathcal{Y}\times\mathcal{Y}}$ respectively as

$$\left(\mathcal{T}_{\mathcal{X}}^{\pi} V_{\mathcal{X}}\right)(xy, x') = -\frac{1}{\eta} \log \sum_{y'} \frac{\pi(x'y'|xy)}{P_{\mathcal{X}}(x'|x)} \exp\left(-\eta\left(c(xy) + \gamma \sum_{x''} P_{\mathcal{X}}(x''|x')V_{\mathcal{X}}(x'y', x'')\right)\right),$$

and

$$\left(\mathcal{T}_{\mathcal{Y}}^{\pi} V_{\mathcal{Y}}\right)(xy, y') = -\frac{1}{\eta} \log \sum_{x'} \frac{\pi(x'y'|xy)}{P_{\mathcal{Y}}(y'|y)} \exp\left(-\eta\left(c(xy) + \gamma \sum_{y''} P_{\mathcal{Y}}(y''|y')V_{\mathcal{Y}}(x'y', y'')\right)\right).$$

Then, for odd rounds, the updates can be calculated by solving the fixed-point equations $V_k = \mathcal{T}_{\mathcal{X}}^{\pi_k} V_k$, defining the shorthand $Q_k(xy, x'y') = c(xy) + \gamma \sum_{x''} P_{\mathcal{X}}(x''|x')V_k(x'y', x'')$, and subsequently updating the transition coupling $\pi_k$ multiplicatively as

$$\pi_{k+1}(x'y'|xy) = \frac{\pi_k(x'y'|xy)\exp\left(-\eta Q_k(xy, x'y')\right)}{\sum_{y''} \pi_k(x'y''|xy)\exp\left(-\eta Q_k(xy, x'y'')\right)} P_{\mathcal{X}}(x'|x). \tag{10}$$

It is easy to verify that this transition coupling satisfies $\sum_{y'} \pi_{k+1}(x'y'|xy) = P(x'|x)$. The updates for even rounds are computed analogously with the roles of $\mathcal{X}$ and $\mathcal{Y}$ swapped. We respectively refer to the fixed-point equations $V_k = \mathcal{T}_{\mathcal{X}}^{\pi_k} V_k$ and $V_k = \mathcal{T}_{\mathcal{Y}}^{\pi_k} V_k$ as the *Bellman–Sinkhorn equations* for $M_{\mathcal{X}}$ and $M_{\mathcal{Y}}$, and the functions $V_k$ and $Q_k$ as *value functions*. The following proposition (proved in Appendix E.1) formally establishes the equivalence between the two update rules.

**Proposition 1.** *Let $\mu_{k+1}$ and $\pi_{k+1}$ be specified for each $k$ as in Equations (9) and (10), respectively. Then, $\mu_{k+1} = \mu^{\pi_{k+1}}$ holds for all $k$.*

### 4.2 Practical implementation

The algorithm described above can be seen as performing online Mirror Sinkhorn updates in each state pair $xy$ with a sequence of cost functions $Q_k$, which are computed via solving the Bellman–Sinkhorn equations. Since $\mathcal{T}_{\mathcal{X}}^{\pi}$ and $\mathcal{T}_{Y}^{\pi}$ are easily seen to be contractive with respect to the supremum norm with contraction factor $\gamma$ (as shown by a standard calculation included in Appendix E.3), these equations can be solved by an adaptation of the classic Value Iteration method of Bellman [1957]. Concretely, we repeatedly apply the Bellman–Sinkhorn operators until the fixed point is reached up to sufficient precision (controlled by the number of update steps $m$). We call the resulting method *Sinkhorn Value Iteration* (SVI), and provide its pseudocode as Algorithm 1.

---

**Algorithm 1:** Sinkhorn Value Iteration

**Input:** $P_{\mathcal{X}}, P_{\mathcal{Y}}, c, \eta, \gamma, K, m$
**Initialise:** $\pi_1 \leftarrow P_{\mathcal{X}} \otimes P_{\mathcal{Y}}$;
**for** $k = 1, ..., K - 1$ **do**
    **if** $k$ *is odd* **then**
        | $V_{\mathcal{X}} \leftarrow (\mathcal{T}_{\mathcal{X}}^{\pi_k})^m V_{\mathcal{X}}$; {$\mathcal{B}_{\mathcal{X}}$ projection.}
    **else**
        | $V_{\mathcal{Y}} \leftarrow (\mathcal{T}_{\mathcal{Y}}^{\pi_k})^m V_{\mathcal{Y}}$; {$\mathcal{B}_{\mathcal{Y}}$ projection.}
    **end**
    $\pi_{k+1} \leftarrow$ **update**$(\pi_k)$;     {Equation 10}
**end**
$\overline{\mu}_K \leftarrow \frac{1}{K} \sum_{k=1}^{K} (\mu^{\pi_k})$;
$\pi_{\text{out}} \leftarrow$ **round**$(\pi_{\overline{\mu}_K})$;
$V^{\pi_{\text{out}}} \leftarrow$ **evaluate**$(\pi_{\text{out}})$;
**Output:** $\pi_{\text{out}}, V^{\pi_{\text{out}}}$     {Final coupling}

---

Notably, while SVI is defined in its abstract form as a sequence of updates in the space of occupancy couplings $\mu_k$, its implementation only works with transition couplings $\pi_k$. The final output of SVI is a transition coupling $\pi_{\text{out}}$, obtained by computing the average $\overline{\mu}_K = \frac{1}{K} \sum_{k=1}^{K} \mu^{\pi_k}$ of all occupancy couplings, computing $\pi_{\overline{\mu}_K}$ and then rounding the result to a valid transition coupling. In particular we apply a simple rounding procedure due to Altschuler et al. [2017] individually on $\pi_{\overline{\mu}_K}(\cdot|xy)$ for each state-pair $xy$—for the full details, see Appendix E.2. Besides $\pi_{\text{out}}$, SVI also outputs an estimate of $V^*$ in the form of the *value function* $V^{\pi_{\text{out}}}$, as defined in Equation (11) in Appendix B.1. This function can be computed efficiently by solving the linear system of Bellman equations $V^{\pi_{\text{out}}}(xy) = c(xy) + \gamma \sum_{x'y'} \pi_{\text{out}}(x'y'|xy)V^{\pi_{\text{out}}}(x'y')$.

A number of small simplifying steps can be made to make the algorithm easier to implement. First, instead of obtaining $\pi_{\overline{\mu}_K}$ via the computationally expensive procedure described above, one can simply run the rounding procedure on the final transition coupling $\pi_K$ and output the result. Second,

while theoretical analysis suggests setting $m = \infty$ in order to make sure that all projection steps are perfect, such exact computation may be unnecessary and inefficient in practice, and thus (much) smaller values can be used instead. Third, for small values of $\eta$ the softmax function used in the definition of the Bellman–Sinkhorn operator can be accurately approximated by an average with respect to $\pi_k(x'y'|xy)/P_\mathcal{X}(x'|x)$, which suggests a simple alternative to the projection steps. This approximates SVI similarly as to how the Mirror Descent Modified Policy Iteration method of Geist et al. [2019] approximates the mirror descent method of Neu et al. [2017] (see also Azar et al., 2012). The resulting method (that we refer to as *Sinkhorn Policy Iteration*, or SPI) is presented in detail along with its theoretical analysis in Appendix D. We study effects of these implementation choices via a sequence of experiments in Section 5.

### 4.3 Convergence guarantees

The following theorem establishes a guarantee on the number of iterations necessary for $V^{\pi_{out}}(xy)$ be an $\varepsilon$-accurate approximation of the transport cost $\mathbb{W}_\gamma(M_\mathcal{X}, M_\mathcal{Y}; c, x, y)$ for any $x, y$.

**Theorem 2.** *Suppose that Sinkhorn Value Iteration is run for $K$ steps with regularization parameter $\eta = \frac{1}{4\|c\|_\infty}\sqrt{\frac{(1-\gamma)^3 \log |\mathcal{X}||\mathcal{Y}|}{K}}$, and initialized with the uniform coupling defined for each $xy, x'y'$ as $\pi_1(x'y'|xy) = \frac{1}{|\mathcal{X}||\mathcal{Y}|}$. Then, for any $x_0 y_0 \in \mathcal{X}\mathcal{Y}$, the output satisfies $V^{\pi_{out}}(x_0 y_0) \leq \mathbb{W}_\gamma(M_\mathcal{X}, M_\mathcal{Y}; c, x_0, y_0) + \varepsilon$ if the number of iterations is at least*

$$K \geq \frac{324 \|c\|_\infty^2 \log |\mathcal{X}||\mathcal{Y}|}{(1-\gamma)^5 \varepsilon^2}.$$

The proof is relegated to Section C, and we present a similar performance guarantee for SPI in Appendix D. Importantly, these guarantees technically only hold when setting $m = \infty$, which is a limitation we discuss in more detail in Section 6. The condition that $\pi_1$ is chosen as the uniform coupling is not necessary and simply made to make the statement easier to state. A more detailed statement of the bound is provided in Appendix C.4.

## 5 Experiments

We have conducted a range of experiments on some simple environments with the purpose of illustrating the numerical properties of our algorithms and some aspects of the distance metrics we studied. Due to space restrictions, we only report a very limited subsample of the results below, and refer the reader to Appendix F for the complete suite[2].

One set of experiments we report here addresses the biggest open question left behind our theory: the effect of the choice of $m$ on the quality of the updates. For this experiment, we use the classic "4-rooms" environment first studied by Sutton et al. [1999], and run both SVI (Algorithm 1) and SPI (Algorithm 2) for a range of different choices of $m$, and a fixed $\gamma = 0.95$. The results of this study are shown in Figure 1. The plots indicate that the estimates produced by both algorithms converge towards the true distance at a rate that is basically unaffected by $m$, and in particular even a value of $m = 1$ remains competitive. This observation is consistent across all of our experiments. Also, the output of SPI appears to converge slightly more slowly towards the optimum in this experiment, but this observation is not entirely consistent and can be likely ascribed to the fact that the learning rate was not optimized to favor either algorithm in this experiment. In most experiments, the two algorithms performed very similarly, up to some small occasional differences.

We have also conducted a number of experiments to illustrate the potential of optimal-transport distances for comparing Markov chains of different sizes and transition functions. In the experiment we show here, we compare two Markov chains illustrated in Figure 2. The first Markov chain $M_\mathcal{X}$ is a simple, nine-state "gridworld" environment, which has its initial state in the upper left corner (denoted as $s_0$) and a reward of $+1$ in the lower left corner (shown in blue). The second, $M_\mathcal{Y}$, is an instance of the 4-rooms environment, where each room is a rotation of the aforementioned small grid. The transition kernels in both environments are uniform distributions over the adjacent cells in the four principal directions. The plot shows the distances between the two chains as a function of the initial state of $M_\mathcal{Y}$, revealing an intuitive pattern of similarities that captures the symmetries of $M_\mathcal{Y}$.

---

[2]The code is available at https://github.com/SergioCalo/SVI

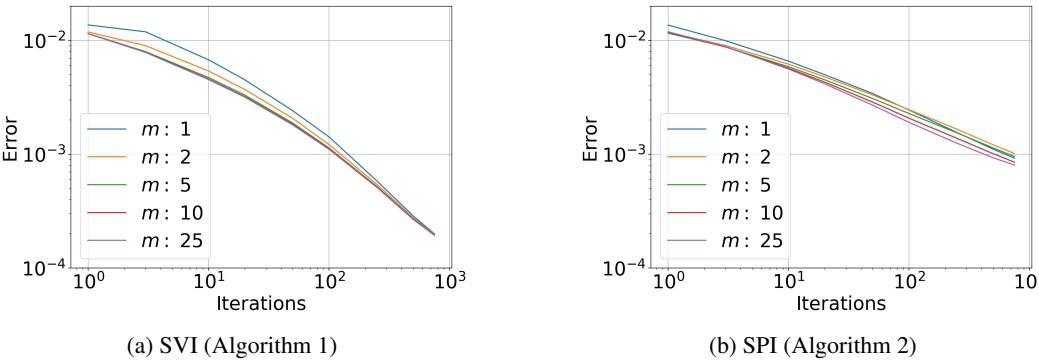

(a) SVI (Algorithm 1)          (b) SPI (Algorithm 2)

Figure 1: Estimated transport cost as a function $k$, for various choices of $m$ and $\eta = 1$.

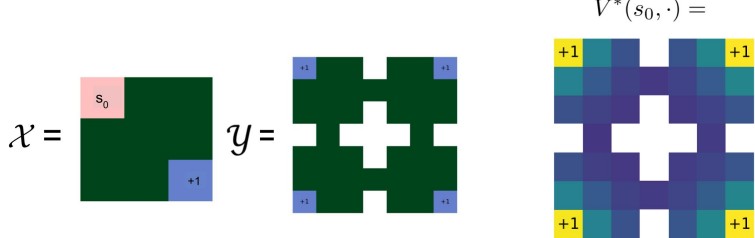

Figure 2: Visual representation of the distances computed between the chains $M_{\mathcal{X}}$ and $M_{\mathcal{Y}}$.

## 6 Discussion

We discuss some further aspects of our framework and results below.

**Representation learning for reinforcement learning.** Among the numerous applications listed in Section 1 and Appendix A, the most interesting for us is using our metrics for representation learning in RL. As mentioned earlier, bisimulation metrics have been extensively used for this purpose in the past. In particular, almost all such work uses bisimulation metrics to compare states within the same MDP and use the resulting similarity metrics for merging states that are at low distance (an approach called "state aggregation"). As our results highlight, this is a rather narrow view of what bisimulation metrics are capable of: they can define similarity metrics between processes that live on potentially different state spaces, which in particular can be used to select representations by minimizing the distance between a high-dimensional process and a set of low-dimensional representations. Curiously, our LP formulation may allow differentiating the distances with respect to the transition kernels, which we believe will be an important property for future developments in representation learning for RL.

**Limitations of the theory.** In their current form, our theoretical guarantees in Theorems 2 and 6 only apply to perfect projection and evaluation steps, corresponding to setting $m = \infty$. We conjecture that this limitation can be addressed with a more careful analysis, and results similar to those of Theorems 2 and 6 can be shown, potentially at the price of a worse dependence on the effective horizon $1/(1 - \gamma)$ [Scherrer et al., 2012, 2015], by making use of the techniques of Geist et al. [2019] and Moulin and Neu [2023] for analyzing regularized dynamic-programming algorithms.

**From dynamic programming to learning from data.** This paper focuses on computing distances between known Markov processes via dynamic-programming-style methods. In the most interesting applications however, the transition kernels are unknown, which requires the development of new tools. We are confident that our framework can serve as a solid basis for such developments, and in particular that one can port many ideas from the field of reinforcement learning that is essentially all about turning dynamic-programming methods into algorithms that can learn from interaction data. Additionally, we believe that our LP formulation in Section 3 makes it much easier to import further ideas from computational optimal transport, and in particular that stochastic optimization methods like those of Genevay et al. [2016] can be adapted to solving our linear programs.

## Acknowledgments.

G. Neu would like to thank Marin Ballu and Quentin Berthet for clarifying a number of details of their analysis of Mirror Sinkhorn, Tristan Brugère for pointing us to their publicly available code, and Anna Korba and Stefan Schrott for kindly helping out with some references about optimal transport between stochastic processes. G. Neu was supported by the European Research Council (ERC) under the European Union's Horizon 2020 research and innovation programme (Grant agreement No. 950180). Anders Jonsson is partially supported by the EU ICT-48 2020 project TAILOR (No. 952215), AGAUR SGR, and the Spanish grant PID2019-108141GB-I00. Javier Segovia-Aguas is supported by MCIN/AEI /10.13039/501100011033 under the Maria de Maeztu Units of Excellence Programme (CEX2021-001195-M).

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

# A Extended discussion of related work

In this appendix we include a discussion of related work that could not be accommodated in the main text due to space limitations.

## A.1 Bisimulation

The concept of bisimulation originated independently in modal logic [van Benthem, 1983], computer science [Park, 1981, Milner, 1989] and set theory [Forti and Honsell, 1983, Aczel, 1988], with firm roots in fixed-point theory. Bisimulation was originally devised as a tool for determining whether or not two processes are behaviorally equivalent in the sense that no test can distinguish between the labels they generate. Being a much less demanding notion of equivalence than isomorphism (which is generally NP-hard to verify), the notion of bisimulation has had significant impact in concurrency theory and formal verification of computer systems, and has become a standard tool for model checking. We refer the reader to the very enjoyable paper of Sangiorgi [2009] for a detailed history of bisimulation and related concepts.

Larsen and Skou [1989] developed a theory of probabilistic bisimulation between stochastic processes. Their approach is based on comparing interpretations of logical formulas on stochastic processes, roughly saying that two processes are probabilistically bisimilar if all formulas acting on the sequence of labels encountered along the corresponding random trajectories follow the same probability distribution. Their definition can be most simply presented when the two processes live on the same state space and follow the same transition kernel, but are initialized at two different states. In this setup, bisimulation reduces to a relation between individual states, which can be described formally as follows. Given a stationary Markov process $M_{\mathcal{X}} = (\mathcal{X}, P_{\mathcal{X}}, \nu_{0,\mathcal{X}})$ as defined in Section 2 and a label function $\mathcal{L} : \mathcal{X} \to \mathbb{R}$, a probabilistic bisimulation $R$ is a relation on $\mathcal{X} \times \mathcal{X}$ that satisfies the following property: two states $x$ and $x'$ are bisimilar (denoted $xRx'$) if and only if $\mathcal{L}(x) = \mathcal{L}(x')$ and for each subset $\mathcal{C}$ in the partition $\mathcal{X} \setminus R$ induced by $R$, it holds that

$$\sum_{x'' \in \mathcal{C}} P_{\mathcal{X}}(x''|x) = \sum_{x'' \in \mathcal{C}} P_{\mathcal{X}}(x''|x').$$

Jonsson and Larsen [1991] define another similarity notion called "satisfaction relation" based on couplings, and prove that probabilistic bisimilarity and satisfaction are equivalent notions of similarity. These works show that bisimulation is indeed an equivalence relation, and that when two processes are initialized from two states within the same equivalence class (i.e., they are *bisimilar*), then they will not only transition to bisimilar states in the next step but will in fact continue to evolve in a way that is indistinguishable based on the labels (in the sense that they will produce the same distribution over sequences of labes).

Giacalone et al. [1990], Desharnais et al. [1999, 2004] and van Breugel and Worrell [2001] relax the restrictive notion of exact probabilistic bisimulation and introduce real-valued pseudometrics that measure the degree of bisimilarity between two states. These notions rely on real-valued labeling functions. Giacalone et al. [1990] gives a notion of $\varepsilon$-bisimulation which relaxes the condition $\mathcal{L}(x) = \mathcal{L}(x')$ in the definition of the hard bisimulation relation given above, and only requires equality to hold up to some $\varepsilon > 0$. Desharnais et al. [1999, 2004] go further and define a genuinely real-valued extension of bisimulation relations by defining bisimulation metrics as described in the main text (and particularly Equation (4)). A fixed-point characterization of these bisimulation metrics was established by van Breugel and Worrell [2001] and Desharnais et al. [2002]. These works show that the resulting distance notion is in fact a pseudometric, and two processes are bisimilar if and only if they are at distance zero. This justifies seeing bisimilarity metrics as "soft" extensions of the binary relation of bisimilarity.

Interestingly, the first bisimulation metrics all make use of concepts from optimal transport in some way or another: Desharnais et al. [1999, 2002] already note that their definition is inspired by the Wasserstein distance (which they call the "Hutchinson metric"), and the fixed-point characterization of van Breugel and Worrell [2001] also make use of this distance to compare transition kernels (cf. Equation 5). Precisely, their definition that we recalled as Equation (4) is admittedly inspired by the Kantorovich dual representation of the Wasserstein distance between probability distributions over metric spaces. To our knowledge, this connection with optimal transport has not been explored further in the literature, and in particular no "primal" counterpart based on couplings has been discovered so far.

Givan et al. [2003] adapt probabilistic bisimulation to Markov decision processes (MDPs), requiring states (or state-action pairs) to have identical rewards and transition probabilities for two MDPs to be bisimilar. Ferns et al. [2004, 2006] introduce bisimulation metrics for MDPs, essentially using the fixed-point characterization of van Breugel and Worrell [2001] as a starting point for their definition. Once again, their definition makes use of the Wasserstein distance between the transition kernels (called Kantorovich metric in the paper), but no deeper connection between bisimulation metrics and optimal transport is discussed.

Castro [2020] defines bisimulation metrics for MDPs with respect to a given policy $\pi$, which thus falls back to the standard definition of bisimulation metrics for Markov chains as studied in the works of Desharnais et al. [1999, 2004] and van Breugel and Worrell [2001]. Kemertas and Jepson [2022b] adjusts the definition of Ferns et al. [2004, 2006] by replacing the Wasserstein distance with the entropy-regularized Wasserstein distance proposed by Cuturi [2013], which allows them to apply the Sinkhorn algorithm to compute bisimulation metrics for MDPs. The resulting approach can be seen to be nearly identical to the methods proposed by O'Connor et al. [2022] and Brugère et al. [2024] for approximately solving the fixed-point equations (3) in the context of optimal transport—see Section A.2 for further discussion of these works.

Chen et al. [2012] study the computational complexity of computing bisimulation metrics. Similar to our work, the authors formulate a linear program that characterizes bisimulation metrics in an equivalent way to other common definitions, though the linear program has one constraint per next-state coupling, which makes their LP intractable as stated. They use their LP formulation as an analytic tool to show the existence of a polynomial-time algorithm to solve the fixed-point equations (5) (which essentially amounts to Bellman's value iteration algorithm implemented in the MDP we describe in Appendix B). Each step of the resulting algorithm solves one optimal-transport problem per state pair via the network simplex algorithm, which is known to be impractical for this purpose in comparison with Sinkhorn-style methods [Cuturi, 2013]. In the context of optimal transport, a closely related linear program has been discovered by Backhoff-Veraguas et al. [2017], whose framework is more general in that it mostly focuses on general (potentially non-Markovian) stochastic processes, but with the limitation that only finite horizons are considered. We discuss further developments on this topic in Section A.2.

Finally, Bian and Abate [2017] show that $\varepsilon$-bisimilar Markov processes generate distributions over finite-length trajectories that are close in total variation distance. Since this distance is a special case of the Wasserstein distance (with the Hamming metric over sequences as ground metric), this result can be seen to establish some relation between OT distances and bisimulation, but the link is rather weak in the sense that no equivalence is shown between the two notions. Indeed, these results only imply that nearly-bisimilar processes generate nearly-identical trajectory distributions, but the reverse implication is not shown to hold.

### A.2 Optimal Transport

Optimal transport [Villani, 2009] studies the problem of transporting mass between two density functions $p$ and $q$, given a cost function that measures the transport distance between any pair of points. In the classic formulation of Kantorovich [1942], the vehicle used to transport mass is a coupling, that is, a joint probability distribution whose marginals equal $p$ and $q$. The problem of finding an optimal coupling that minimizes the total transport distance can be formulated as a linear program. Historically, the resulting LPs have been solved via standard solvers like the network simplex method or interior-point methods, which lead to algorithms with polynomial runtime guarantees but rather poor empirical performance. Cuturi [2013] successfully advocated for adding entropy regularization to the standard LP objective, which enabled algorithms that are orders of magnitude faster than previously proposed methods.

In this work we consider a problem of optimal transport between stochastic processes with a temporal dimension. This topic has recently started to receive attention in the OT literature, mostly focusing on stochastic processes with finite horizon [Pflug and Pichler, 2012, Backhoff-Veraguas et al., 2017, Lassalle, 2018]. In this setting (often called "adapted transport", "causal transport", or "bicausal transport"), the problem is to transport mass between joint distributions of sequences of elements, which can be formulated as an optimization problem over the set of causal couplings (i.e., the set of couplings over joint distributions over sequences that respect the temporal order inherent in the process). Backhoff-Veraguas et al. [2017] have observed that, due to the linearity of the

causality constraints, this optimization problem can be phrased as a linear program, which however is infinite-dimensional and thus intractable to solve directly. They complement this view by providing dynamic-programming principles for characterizing the structure of the optimal coupling, which, in the special case of Markov processes, boils down to the finite-horizon version of the fixed point equations (3). This development essentially mirrors the LP formulation and dynamic-programming principles put forth by Chen et al. [2012] in the context of computing bisimulation metrics (cf. the discussion in Section A.1).

Still on the front of computing optimal transport distances, a notable contribution is due to Eckstein and Pammer [2024], who propose and analyze a version of Sinkhorn's algorithm for optimal transport on the space of stochastic processes. When specialized to Markov processes, their algorithm can be seen to be very closely related to ours, the technical explanation being that in finite-horizon Markov processes the entropy of path distributions that they use as regularization can be seen to be equal to the conditional entropy that our method uses for the same purpose. The resulting algorithm performs iterative Bregman projections via backward recursion over the finite time horizon, with computational steps that are essentially identical to applying our Bellman–Sinkhorn operators. That said, their analysis relies very heavily on the finite-horizon structure of the problem and as such it is not applicable in our considerably more challenging infinite-horizon problem setting.

The more recent works of Moulos [2021], O'Connor et al. [2022], Bayraktar and Han [2023] and Brugère et al. [2024] have investigated optimal-transport distances between infinite-horizon Markov chains. O'Connor et al. [2022] considered the undiscounted version of our problem and proposed to compute optimal transition couplings via an adaptation of approximate policy iteration (cf. Scherrer 2013) to an appropriately adjusted version of the MDP we describe in Appendix B. Their key algorithmic idea is approximating the greedy policy update steps by running Sinkhorn's algorithm for each pair of states. Essentially the same idea was used by Brugère et al. [2024] to solve the discounted problem that is the subject of the present paper, with the difference that their method takes approximate value iteration as its starting point. Both of these approaches are closely related to the alternative fixed-point definition of bisimulation metrics using Sinkhorn divergences due to Kemertas and Jepson [2022b], as mentioned in Section A.1. While these approaches are nearly as effective as our Sinkhorn Value Iteration method in practice, their black-box use of Sinkhorn's algorithm make them difficult to analyze theoretically, and difficult to build further theory on.

To wrap up, let us mention some results that in a sense have already foreshadowed our observation about the relation of OT distances and bisimulation metrics. First, we note that Yi et al. [2021], Brugère et al. [2024] proposed to study optimal transport distances of Markov chains defined over graphs as a means of studying the similarity of the underlying graphs. The purpose of these works was to define a notion of distance that is less demanding than isomorphism, but is still grounded in fundamental theory and can be computed effectively—which is precisely the reason that the notion of bisimulation was originally introduced in the 1980s in the context of formal verification by Park [1981] and Milner [1989]. Finally, the work of Backhoff-Veraguas et al. [2020] has established that "all adapted topologies are equal" on the space of laws of stochastic processes, understood in the sense that a large number of topologies (including the one induced by optimal-transport metrics defined in terms of bicausal couplings) are in fact identical. While one may argue with their sweeping claim that *all* such topologies are equal, it may not be surprising in light of their results that the topology induced by bisimulation metrics is also identical to these well-studied topologies (as revealed to be true by our observations in this paper).

# B Optimal Transport as a Markov Decision Process

Consider the two Markov processes $M_{\mathcal{X}}$ and $M_{\mathcal{Y}}$ with initial states $x_0$ and $y_0$, respectively. To compute the optimal transport cost $\mathbb{W}_\gamma(M_{\mathcal{X}}, M_{\mathcal{Y}}; c, x_0, y_0)$ and the optimal transition coupling $\pi^*$, we can introduce a Markov decision process

$\mathcal{M} = (\mathcal{X}\mathcal{Y}, \mathcal{A}, q, \gamma, c, \nu_0)$ where:

- $\mathcal{X}\mathcal{Y}$ is the state space, defined as the set of joint states of $M_{\mathcal{X}}$ and $M_{\mathcal{Y}}$,
- $\mathcal{A}(xy) = \Pi_{xy}$ is the set of applicable actions in state $xy \in \mathcal{X}\mathcal{Y}$, corresponding to the set of valid couplings of $P_{\mathcal{X}}(\cdot|x)$ and $P_{\mathcal{Y}}(\cdot|y)$,
- $q(\cdot|xy, a) = a$ is the transition probability distribution, which is fully determined by the action $a \in \Pi_{xy}$,
- $\gamma$ is the discount factor,
- $c : \mathcal{X}\mathcal{Y} \to [0, \infty)$ is the cost function that maps joint states to positive real numbers,
- $\nu_0 = \delta_{x_0 y_0}$ is the initial state distribution.

The objective of the agent in this MDP is to select its sequence of actions $A_0, A_1, \ldots$ in a way that minimizes the total discounted cost $\mathbb{E}\left[\sum_{t=0}^\infty \gamma^t c(X_t, Y_t)\right]$, where each state pair is drawn according to the action taken by the agent as $(X_t, Y_t) \sim A_t$. The sequence of actions is generated by a sequence of history dependent policies $\pi_t \in \Pi_{\text{HD}} : \mathcal{H}_t \to \Delta_{\mathcal{A}(X_t, Y_t)}$ where $\mathcal{H}_t = (X_0, Y_0, \ldots, X_t, Y_t)$. A first remark is that we can restrict ourselves to deterministic policies. Indeed, the action set is convex at every time step and both the reward and the transition probability distributions are linear in the action. A second remark is that bicausal couplings correspond exactly to history-dependent deterministic policies. Indeed, by bicausality, the bicausal coupling $M_{\mathcal{X}\mathcal{Y}}$ is generated by the policy $M_{\mathcal{X}\mathcal{Y}}(x_n, y_n | \bar{x}_{n-1} \bar{y}_{n-1})$. We will denote this policy by $\pi_{M_{\mathcal{X}\mathcal{Y}}}$.

Of special interest are stationary deterministic (or Markovian) policies of the form $\pi : \mathcal{X}\mathcal{Y} \to \mathcal{A}$, mapping joint states $(X_t, Y_t)$ to actions in $\mathcal{A}(X_t, Y_t)$ as $A_t = \pi(X_t, Y_t)$. Such policies correspond exactly with transition couplings as defined in the main text as mappings $\pi : \mathcal{X}\mathcal{Y} \to \Delta_{\mathcal{X}\mathcal{Y}}$ of the same type. Accordingly, we will sometimes write $\pi(\cdot|xy)$ to refer to the distribution $\pi(xy) \in \mathcal{A}(xy)$ below.

## B.1 Value functions, optimal policies, and sufficiency of transition couplings

Each policy $\pi$ induces a value function $V^\pi$, defined in each state $xy \in \mathcal{X}\mathcal{Y}$ as

$$V^\pi(xy) = \mathbb{E}_\pi\left[\sum_{t=0}^\infty \gamma^t c(X_t Y_t) \,\middle|\, X_0 Y_0 = xy\right], \tag{11}$$

where the expectation is taken with respect to the stochastic process induced by the policy $\pi$. Here, $X_t$ and $Y_t$ are random variables representing the state of the two processes at time $t$. In particular, we have that for any bicausal coupling $M_{\mathcal{X}\mathcal{Y}} \in \Pi_{\text{bc}}$,

$$\int c_\gamma(\bar{X}, \bar{Y}) dM_{\bar{X}, \bar{Y}} = \int \sum_{t=0}^\infty \gamma^t c(X_t, Y_t) dM_{\mathcal{X}\mathcal{Y}}(\bar{X}, \bar{Y})$$

$$= \mathbb{E}_{\pi_{M_{\mathcal{X}\mathcal{Y}}}}\left[\sum_{t=0}^\infty \gamma^t c(X_t, Y_t) \,\middle|\, X_0 Y_0 = x_0 y_0\right]$$

$$= V^{\pi_{M_{\mathcal{X}\mathcal{Y}}}}(x_0, y_0).$$

And as a result we can relate the optimal transport cost to the optimal value function of the MDP

$$\mathbb{W}_\gamma(M_{\mathcal{X}}, M_{\mathcal{Y}}; c, x_0, y_0) = \inf_{M_{\mathcal{X}\mathcal{Y}} \in \Pi_{\text{bc}}} \int c_\gamma(\overline{X}, \overline{Y}) \, dM_{\mathcal{X}\mathcal{Y}}(\overline{X}, \overline{Y})$$

$$= \inf_{M_{\mathcal{X}\mathcal{Y}} \in \Pi_{\text{bc}}} V^{\pi_{M_{\mathcal{X}\mathcal{Y}}}}(x_0 y_0)$$

$$= \inf_{\pi \in \Pi_{\text{HD}}} V^\pi(x_0 y_0).$$

Building on classic results of MDP theory, it can be shown that there exists an optimal Markovian policy $\pi^*$ whose value function $V^* = V^{\pi^*}$ satisfies $V^*(xy) \leq V^\pi(xy)$ for all policies $\pi$ and joint states $xy$, and said optimal value function $V^*$ satisfies the Bellman optimality equations

$$V^*(xy) = \mathcal{T}V^*(xy),$$

where $\mathcal{T}$ is the Bellman operator acting on a function $V \in \mathbb{R}^{\mathcal{XY}}$ as

$$\mathcal{T}V(xy) = c(xy) + \gamma \inf_{p \in \Pi_{xy}} \sum_{x'y'} p(x'y')V(x'y') \quad (\forall xy).$$

This is precisely the set of equations in Equation (3). Since $V^*$ is optimal, $V^*(xy)$ equals the optimal transport cost $\mathbb{W}_\gamma(M_\mathcal{X}, M_\mathcal{Y}; c, x, y)$ for each state $xy$. An optimal policy $\pi^*$ achieves the infimum in each state $xy$, with associated value function $V^{\pi^*} = V^*$. These claims are summarized in the following theorem, stated in nearly identical form by Moulos [2021].

**Theorem 3** (cf. Theorem 1 in Moulos 2021). *Under the above conditions, the following hold:*

- *There exists an optimal Markovian transition coupling $\pi^*$ such that for any policy $\pi \in \Pi_{HD}$ and any joint states $xy$, $V^{\pi^*}(xy) \leq V^\pi(xy)$.*

- *The value function of $\pi^*$ satisfies $\mathbb{W}_\gamma(M_\mathcal{X}, M_\mathcal{Y}; c, x, y) = V^{\pi^*}(xy)$.*

- *There exists a unique solution to the Bellman optimality equation (3) denoted $V^* \in \mathbb{R}^{\mathcal{XY}}$.*

- *We have that $V^* = V^{\pi^*}$.*

The above theorem justifies considering Markovian transition couplings when computing the optimal transport cost $\mathbb{W}_\gamma(M_\mathcal{X}, M_\mathcal{Y}; c, x_0, y_0)$.

*Proof.* One can straightforwardly check that our setting is an instance of optimal control problems with additive cost functional studied in Bertsekas [1977]. In particular, the contraction assumption (Assumption C in the paper mentioned above) is satisfied. We define pointwise $V^*(xy) = \inf_{\pi \in \Pi_{HD}} V^\pi(xy)$, and Proposition 1 of Bertsekas [1977] shows that $V^*$ is the unique solution to the Bellman optimality equation.

Now, for the existence of an optimal stationary policy or transition coupling in our terminology, an additional technical condition on the action set must be carefully verified. For every $xy \in \mathcal{XY}$, $\ell \in [0, \infty)$ and $k$, we define the set

$$U_k(xy, \lambda) = \left\{ a \in \mathcal{A}(xy) : c(x, y) + \gamma \int \mathcal{T}^k V(x'y') \mathrm{d}a(x'y') \leq \lambda \right\}$$

This set is compact as the intersection of the compact set $\mathcal{A}(xy)$ with a closed set, the preimage of a closed set by a continuous application. Knowing that $U_k(xy, \lambda)$ is compact, we can then apply Proposition 14 of Bertsekas [1977] and that gives us the existence of an optimal Markovian transition coupling $\pi^*$ that satisfies $V^{\pi^*} = V^*$. In particular, for any policy $\pi \in \Pi_{HD}$ and joint state $xy$, we have that $V^{\pi^*}(xy) = V^*(xy) \leq V^\pi(xy)$ by definition of $V^*$. $\qquad\square$

### B.2 Occupancy measures and occupancy couplings

In a finite Markov decision process, occupancy measures express the discounted number of times that a given state and action are visited on expectation by the controlled stochastic process. This notion is not meaningfully applicable in the MDP formulation of our optimal-transport problem, given that the action space is infinite. However, the closely related notion of occupancy coupling can be seen to play a similar role in that it allows expressing the total-discounted-cost objective as a linear function, and that the set of valid occupancy couplings can be fully characterized in terms of a finite number of linear constraints. In what follows, we prove this latter key property of occupancy couplings, stated as Lemma 1 in the main text.

*Proof of Lemma 1.* We begin by showing that the occupancy coupling $\mu^\pi$ associated with any transition coupling $\pi \in \Pi_{\mathcal{XY}}$ satisfies the following system of equations (sometimes called the "Bellman

flow equations"):

$$\mu^\pi(xy, x'y') = \pi(x'y'|xy)\left(\gamma \sum_{x''y''} \mu^\pi(x''y'', xy) + (1-\gamma)\nu_0(xy)\right). \qquad (12)$$

Indeed, this can be shown to follow from the definition of occupancy couplings as

$$\mu^\pi(xy, x'y') = (1-\gamma)\sum_{t=0}^{\infty} \gamma^t \mathbb{P}_\pi\left[X_t Y_t = xy, X_{t+1}Y_{t+1} = x'y'\right]$$

$$= (1-\gamma)\sum_{t=0}^{\infty} \gamma^t \pi(x'y'|xy)\mathbb{P}_\pi\left[X_t Y_t = xy\right]$$

$$= \pi(x'y'|xy)\left((1-\gamma)\nu_0(xy) + (1-\gamma)\sum_{t=1}^{\infty} \gamma^t \mathbb{P}_\pi\left[X_t Y_t = xy\right]\right)$$

$$= \pi(x'y'|xy)\left((1-\gamma)\nu_0(xy) + \gamma \sum_{x''y''}(1-\gamma)\sum_{t=1}^{\infty} \gamma^{t-1}\mathbb{P}_\pi\left[X_{t-1}Y_{t-1} = x''y'', X_t Y_t = xy\right]\right)$$

$$= \pi(x'y'|xy)\left((1-\gamma)\nu_0(xy) + \gamma \sum_{x''y''} \mu^\pi(x''y'', xy)\right),$$

where in the first step we used the stationarity of the transition coupling $\pi$, then the definition of $\nu_0$, followed by the law of total probability, and finally the stationarity of the Markov chain that allowed us to recognize $\mu^\pi(x''y'', xy)$ in the last step. Now, summing both sides of Equation (12) for all $x'y'$, we can confirm that $\mu^\pi$ indeed satisfies Equation (6). Furthermore, summing the two sides of Equation (12) over all $x'$, we get

$$\sum_{x'}\mu^\pi(xy, x'y') = \sum_{x'}\pi(x'y'|xy)\left((1-\gamma)\nu_0(xy) + \gamma\sum_{x''y''}\mu^\pi(x''y'', xy)\right)$$

$$= \sum_{x'}\pi(x'y'|xy)\sum_{x''y''}\mu^\pi(xy, x''y'')$$

$$= P_{\mathcal{Y}}(y'|y)\sum_{x''y''}\mu^\pi(xy, x''y''),$$

where the first step in the second line follows from Equation 6, and the 4th line comes from the fact that $\pi(\cdot|xy)$ is a coupling of $P_{\mathcal{X}}(\cdot|x)$ and $P_{\mathcal{Y}}(\cdot|y)$. This verifies that $\mu^\pi$ satisfies Equation (8), and the same reasoning can be used to verify that it also satisfies Equation 7.

Conversely, suppose that $\mu \in \mathbb{R}_+^{\mathcal{XY}\times\mathcal{XY}}$ satisfies Equations (6), (7), and (8). Define $\nu_\mu(xy) = \sum_{x'y'}\mu(xy, x'y')$ and let

$$\pi_\mu(x'y'|xy) = \begin{cases} \frac{\mu(xy,x'y')}{\nu_\mu(xy)} & \text{if } \nu_\mu(xy) \neq 0, \\ P_{\mathcal{X}}(x'|x)P_{\mathcal{Y}}(y'|y) & \text{otherwise}. \end{cases}$$

We will verify that $\pi_\mu$ defines a valid Markovian coupling and that $\mu$ is the state action occupancy measure of $\pi_\mu$. If $\nu_\mu(xy) = 0$, then $\sum_{x'}\pi_\mu(x'y'|xy) = P_{\mathcal{Y}}(y'|y)\sum_{x'}P_{\mathcal{X}}(x'|x) = P_{\mathcal{Y}}(y'|y)$. If $\nu_\mu(xy) \neq 0$, then we have

$$\sum_{x'}\pi_\mu(x'y'|xy) = \frac{1}{\nu_\mu(xy)}\sum_{x'}\mu(xy, x'y') = \frac{1}{\nu_\mu(xy)}\sum_{x'y''}\mu(xy, x'y'')P_{\mathcal{Y}}(y'|y)$$

$$= \frac{1}{\nu_\mu(xy)}\nu_\mu(xy)P_{\mathcal{Y}}(y'|y) = P_{\mathcal{Y}}(y'|y),$$

where we have used that $\mu$ satisfies the constraint of Equation (8). In any case, we have that $\sum_{x'}\pi_\mu(x'y'|xy) = P_{\mathcal{Y}}(y'|y)$. By symmetry, using that $\mu$ satisfies Equation (7), we also have

$\sum_{y'} \mu(x'y'|xy) = P_{\mathcal{X}}(x'|x)$. Hence we have demonstrated that $\pi_\mu$ is a valid Markovian coupling of $P_{\mathcal{X}}$ and $P_{\mathcal{Y}}$. To proceed, observe that the occupancy coupling associated with $\pi_\mu$ satisfies

$$\sum_{x'y'} \mu^{\pi_\mu}(xy, x'y') = \gamma \sum_{x''y''} \mu^{\pi_\mu}(x''y'', xy) + (1-\gamma)\nu_0(xy).$$

We will verify that $\mu^{\pi_\mu} = \mu$ by showing that this system of equations has a unique solution. In order to see this, let us recall the definition of $\nu_\mu$ and reorder Equation (6) as

$$(1-\gamma)\nu_0(xy) = \nu_\mu(xy) - \gamma \sum_{x''y''} \nu_\mu(x''y'')\pi_\mu(xy|x''y'').$$

Introducing the matrix $Z \in \mathbb{R}^{|\mathcal{X}||\mathcal{Y}| \times |\mathcal{X}||\mathcal{Y}|}$ with entries $Z(xy, x'y') = \pi_\mu(x'y'|xy)$ and representing the functions $\nu_0$ and $\nu_\mu$ in matrix form, this system of equations can be written as

$$(1-\gamma)\nu_0 = (I - \gamma Z)\nu_\mu.$$

Now, thanks to the Perron–Frobenius theorem, the stochastic matrix $Z$ has spectral radius 1, and thus $(I - \gamma Z)$ is invertible, meaning that there is a unique solution $\nu_\mu = (1-\gamma)(I - \gamma Z)^{-1}\nu_0$. This in turn implies that $\mu = \mu^{\pi_\mu}$, thus verifying that $\mu$ is indeed an occupancy coupling induced by a valid transition coupling $\pi_\mu$ if it satisfies Equations (6)-(8). This concludes the proof. $\qquad \square$

# C  The proof of Theorem 2

The proof is composed of two main parts: showing a bound on the *regret* of the iterates $\mu_1, \mu_2, \ldots, \mu_K$, and then accounting for the errors incurred when rounding the average iterate $\overline{\mu}_K = \frac{1}{K} \sum_{k=1}^{K} \mu_k$ to a feasible occupancy coupling. We start by stating some general results that will be useful throughout the analysis, and then study the two sources of error mentioned above separately. For any $\mu \in \mathbb{R}^{\mathcal{XY} \times \mathcal{XY}}$, we define $E\mu(xy) = \sum_{x',y'} \mu(x'y', xy)$, $E^\mathsf{T}\mu(xy) = \sum_{x',y'} \mu(xy, x'y')$ and

$$
\pi_\mu(x'y'|xy) = \begin{cases} \frac{\mu(xy, x'y')}{E^\mathsf{T}\mu(xy)} & \text{if } E^\mathsf{T}\mu(xy) \neq 0, \\ P_\mathcal{X}(x'|x)P_\mathcal{Y}(y'|y) & \text{otherwise.} \end{cases}
$$

In particular, we always have that $\mu(xy, x'y') = E^\mathsf{T}\mu(xy)\pi_\mu(x'y'|xy)$. Furthermore, for any $\pi : \mathcal{XY} \to \Delta_{\mathcal{XY}}$, we will denote by $\tilde{\pi} = \rho(\pi)$ the rounded transition coupling obtained by adapting the rounding procedure of Altschuler et al. [2017], and we will let $\rho(\mu)$ denote the corresponding occupancy coupling $\mu^{\tilde{\pi}_\mu}$ (cf. Section E.2 for the description and analysis of this method). In particular, $\tilde{\pi}$ will always be a valid transition coupling associated with $P_\mathcal{X}, P_\mathcal{Y}$. Finally we define $\pi_\text{out} = \rho(\pi_{\overline{\mu}_K})$ and $\mu_\text{out} = \mu^{\pi_\text{out}}$, and we let $\mu^*$ be an optimal occupancy coupling achieving the minimum in the problem formulation of Theorem 1.

With these notations, we decompose the overall error of the output as

$$
\langle \mu_\text{out} - \mu^*, c \rangle = \langle \mu_\text{out} - \overline{\mu}_K, c \rangle + \langle \overline{\mu}_K - \mu^*, c \rangle = \langle \mu_\text{out} - \overline{\mu}_K, c \rangle + \frac{1}{K} \sum_{k=1}^{K} \langle \mu_k - \mu^*, c \rangle .
$$

The first sum on the right-hand side corresponds to the rounding error, and the second one to the so-called *regret* of the sequence of iterates $\mu_k$. This latter sum can be controlled by adapting arguments from the classic analysis of mirror-descent methods [Nemirovski and Yudin, 1983, Beck and Teboulle, 2003], with some ideas adopted from the Mirror Sinkhorn analysis of Ballu and Berthet [2023] that will also come in handy for analyzing the rounding errors. We state these tools first below, and then analyze the two terms in the above decomposition separately.

## C.1  General tools

We begin with a version of the classic "three-point identity" for mirror-descent methods (e.g., Lemma 4.1 of Beck and Teboulle, 2003) adapted to our specific setting that involves alternating projections to the sets $\mathcal{B}_\mathcal{X}$ and $\mathcal{B}_\mathcal{Y}$. The result is similar to Lemma A.3 of Ballu and Berthet [2023], which we reprove here with a more standard methodology (as used for proving, e.g., Theorem 28.4 of Lattimore and Szepesvári, 2020).

**Lemma 2.** *Let $\mu^* \in \mathcal{B}_\mathcal{X} \cap \mathcal{B}_\mathcal{Y}$ be arbitrary. Then,*

$$
\langle \mu_{k+1} - \mu^*, c \rangle \leq \frac{\mathcal{H}(\mu^* \| \mu_k) - \mathcal{H}(\mu^* \| \mu_{k+1}) - \mathcal{H}(\mu_{k+1} \| \mu_k)}{\eta} .
$$

*Proof.* We first recall that $\mathcal{H}$ is the Bregman divergence associated with the conditional entropy function $\mathcal{C}(\mu) = \sum_{xy, x'y'} \mu(xy, x'y') \log \frac{\mu(xy, x'y')}{\sum_{x''y''} \mu(xy, x''y'')}$ (cf. Appendix A.1 in Neu et al. 2017). For the actual proof, let us consider the case of $k$ odd when $\mu_{k+1} \in \mathcal{B}_\mathcal{X}$. Then, by definition, we have that $\mu_{k+1}$ is the minimizer of $\Psi_{k+1}(\mu) = \eta \langle \mu, c \rangle + \mathcal{H}(\mu \| \mu_k)$ on this set. Noticing that the gradient of $\Psi_{k+1}$ at $\mu$ is written as $\nabla \Psi_{k+1}(\mu) = \eta c + \nabla \mathcal{C}(\mu) - \nabla \mathcal{C}(\mu_k)$, the first-order optimality condition over the convex set $\mathcal{B}_\mathcal{X}$ implies that the following inequality holds for any $\mu \in \mathcal{B}_\mathcal{X}$:

$$
\langle \eta c + \nabla \mathcal{C}(\mu_{k+1}) - \nabla \mathcal{C}(\mu_k), \mu - \mu_{k+1} \rangle \geq 0.
$$

In particular, using this result for $\mu = \mu^*$ (which is indeed in $\mathcal{B}_\mathcal{X}$), the claim follows from using the standard three-point identity of Bregman divergences that states

$$
\langle \nabla \mathcal{C}(\mu_{k+1}) - \nabla \mathcal{C}(\mu_k), \mu^* - \mu_{k+1} \rangle = \mathcal{H}(\mu^* \| \mu_k) - \mathcal{H}(\mu^* \| \mu_{k+1}) - \mathcal{H}(\mu_{k+1} \| \mu_k) .
$$

Repeating the same argument for even rounds (and noticing that the comparator also satisfies $\mu^* \in \mathcal{B}_\mathcal{Y}$ as needed for that case) completes the proof. □

The following standard lemma will also be useful for studying various notions of distances between occupancy couplings: the total variation distance $\|\mu - \mu'\|_1 = \sum_{xy,x'y'} |\mu(xy, x'y') - \mu'(xy, x'y')|$, the relative entropy $\mathcal{D}(\mu\|\mu') = \sum_{xy,x'y'} \mu(xy, x'y') \log \frac{\mu(xy,x'y')}{\mu'(xy,x'y')}$, and the conditional relative entropy introduced earlier.

**Lemma 3.** *For any two occupancy couplings $\mu$ and $\mu'$, we have*

$$\frac{1}{2} \|\mu - \mu'\|_1^2 \le \mathcal{D}(\mu\|\mu') \le \frac{\mathcal{H}(\mu\|\mu')}{1 - \gamma}.$$

*Proof.* The first inequality is Pinsker's. For proving the second inequality, we let $\nu = E^T \mu$ and $\nu' = E^T \mu'$ be the state occupancies associated with $\mu$ and $\mu'$, respectively. Then, we write the following:

$$\mathcal{D}(\mu\|\mu') = \mathcal{D}(\nu\|\nu') + \mathcal{H}(\mu\|\mu')$$
$$\text{(by the chain rule of the relative entropy)}$$
$$= \mathcal{D}((1-\gamma)\nu_0 + \gamma E\mu\|(1-\gamma)\nu_0 + \gamma E\mu') + \mathcal{H}(\mu\|\mu')$$
$$\text{(using that } \mu \text{ and } \mu' \text{ are valid occupancy couplings)}$$
$$\le (1-\gamma)\mathcal{D}(\nu_0\|\nu_0) + \gamma\mathcal{D}(E\mu\|E\mu') + \mathcal{H}(\mu\|\mu')$$
$$\text{(using the joint convexity of the relative entropy)}$$
$$\le \gamma\mathcal{D}(\mu\|\mu') + \mathcal{H}(\mu\|\mu'),$$

where the final step follows from using the data-processing inequality for the relative entropy. Reordering the terms concludes the proof. $\square$

### C.2 Constraint violations

Let us begin with some definitions. First of all we introduce a quantity that measures the extent to which an occupancy coupling $\mu$ violates the transition coherence constraints. Specifically, we will measure the violations of the $\mathcal{X}$-constraints by

$$\delta_{\mathcal{X}}(\mu) = \sum_{xyx'} \left| \nu_\mu(xy) P_{\mathcal{X}}(x'|x) - \sum_{y'} \mu(xy, x'y') \right|$$

and the violations of the $\mathcal{Y}$-constraints by

$$\delta_{\mathcal{Y}}(\mu) = \sum_{xyx'} \left| \nu_\mu(xy) P_{\mathcal{Y}}(y'|y) - \sum_{x'} \mu(xy, x'y') \right|,$$

and the overall constraint violations will be written as

$$\delta(\mu) = \delta_{\mathcal{X}}(\mu) + \delta_{\mathcal{Y}}(\mu).$$

Note that we have $\delta_{\mathcal{X}}(\mu_k) = 0$ for odd rounds and $\delta_{\mathcal{Y}}(\mu_k) = 0$ for even rounds by definition of the updates. We also define the rounding error associated with an occupancy coupling $\mu$ as the average total variation distance between the transition coupling $\pi_\mu$ and its rounded counterpart $\widetilde{\pi}_\mu = \rho(\pi_\mu)$:

$$\Delta(\mu) = \sum_{xy,x'y'} \mu(xy, x'y') \|\widetilde{\pi}_\mu(\cdot|xy) - \pi_\mu(\cdot|xy))\|_1.$$

The first statement establishes a link between the quality of an occupancy coupling and the occupancy coupling obtained after rounding the transition coupling to satisfy the constraints.

**Lemma 4.** *For any $\mu$ satisfying Equation* (6)*, we have*

$$\langle \rho(\mu) - \mu, c \rangle \le \|c\|_\infty \frac{\Delta(\mu)}{(1 - \gamma)}.$$

*Proof.* Let $\tilde{\pi}_\mu = \rho(\pi_\mu)$, $\tilde{\mu} = \rho(\mu)$, $\nu_\mu = E^\top \mu$, and $\nu_{\tilde{\mu}} = E^T \tilde{\mu}$. Furthermore, let us define the shorthand notation $\nu \circ \pi$ to denote the composition of the state-pair distribution $\nu$ with the transition coupling $\pi$ as $(\nu \circ \pi)(xy, x'y') = \pi(x'y'|xy)\nu(xy)$. Then, we have

$$
\begin{aligned}
\|\tilde{\mu} - \mu\|_1 &= \|\nu_{\tilde{\mu}} \circ \tilde{\pi}_\mu - \nu_\mu \circ \pi_\mu\|_1 = \|\nu_{\tilde{\mu}} \circ \tilde{\pi}_\mu + \nu_\mu \circ \tilde{\pi}_\mu - \nu_\mu \circ \tilde{\pi}_\mu - \nu_\mu \circ \pi_\mu\|_1 \\
&= \|(\nu_{\tilde{\mu}} - \nu_\mu) \circ \tilde{\pi}_\mu + \nu_\mu \circ (\tilde{\pi}_\mu - \pi_\mu)\|_1 \\
&\leq \|(\nu_{\tilde{\mu}} - \nu_\mu) \circ \tilde{\pi}_\mu\|_1 + \|\nu_\mu \circ (\tilde{\pi}_\mu - \pi_\mu)\|_1 \\
&\qquad \text{(using the triangle inequality)} \\
&= \|\nu_{\tilde{\mu}} - \nu_\mu\|_1 + \Delta(\mu) \\
&\qquad \text{(using the definition of } \Delta) \\
&= \gamma \|E\tilde{\mu} - E\mu\|_1 + \Delta(\mu) \\
&\qquad \text{(using } \nu_\mu = (1 - \gamma)\nu_0 + \gamma E\mu) \\
&\leq \gamma \|\tilde{\mu} - \mu\|_1 + \Delta(\mu),
\end{aligned}
$$

where in the last step we used that $E$ is non-expansive with respect to the $\ell_1$-norm. Reordering gives

$$
\|\tilde{\mu} - \mu\|_1 \leq \frac{\Delta(\mu)}{1 - \gamma},
$$

and putting everything together proves the claim of the lemma. $\qquad\square$

The next result relates the rounding errors to the constraint violations.

**Lemma 5.** *For any $\mu \in \Delta_{\mathcal{XY} \times \mathcal{XY}}$, we have*

$$
\frac{1}{2}\Delta(\mu) \leq \delta(\mu)
$$

The proof of this result builds on the error analysis of the rounding procedure of Altschuler et al. [2017], and can be found in Appendix E.2. Finally, the last technical lemma (inspired by Lemma A.4 of Ballu and Berthet, 2023) bounds the rounding errors in terms of the change rate of the occupancy couplings.

**Lemma 6.** *For any $k \geq 1$,*

$$
\delta(\mu_k) \leq 2\min(\|\mu_k - \mu_{k+1}\|_1, \|\mu_k - \mu_{k-1}\|_1).
$$

*Proof.* We study the case where the $k$th update is a $\mathcal{B}_\mathcal{Y}$ projection. For this proof, it will be convenient to introduce the following notation. We define $\mathcal{I}_\mathcal{X} : \mathbb{R}^{\mathcal{XY} \times \mathcal{XY}} \to \mathbb{R}^{\mathcal{XY} \times \mathcal{X}}$ and (with some abuse of notation), $P_\mathcal{X} : \mathbb{R}^{\mathcal{XY} \times \mathcal{XY}} \to \mathbb{R}^{\mathcal{XY} \times \mathcal{X}}$ as the linear operators that respectively act on $\mu$ via the assignment $(\mathcal{I}_\mathcal{X}\mu)(xy, x') = \sum_{y'} \mu(xy, x'y')$ and $(P_\mathcal{X}\mu)(xy, x') = \sum_{x'',y''} \mu(xy, x''y'')P_\mathcal{X}(x'|x)$. This allows us to write $\delta_\mathcal{X}(\mu) = \|(\mathcal{I}_\mathcal{X} - \mathcal{P}_X)\mu\|_1$, so that we have the expression

$$
\delta(\mu_k) = \delta_\mathcal{X}(\mu_k) + \delta_\mathcal{Y}(\mu_k) = \|\mathcal{I}_\mathcal{X}\mu_k - P_\mathcal{X}\mu_k\|_1,
$$

where we have also used $\delta_\mathcal{Y}(\mu_k) = 0$ that holds thanks to the fact that $k$ is even. Moreover, the $(k + 1)$st update is a $\mathcal{B}_\mathcal{X}$-projection and thus we have $\mathcal{I}_\mathcal{X}\mu_{k+1} = P_\mathcal{X}\mu_{k+1}$. Hence,

$$
\begin{aligned}
\delta(\mu_k) &= \|\mathcal{I}_\mathcal{X}\mu_k - P_\mathcal{X}\mu_k\|_1 \\
&= \|\mathcal{I}_\mathcal{X}\mu_k - \mathcal{I}_\mathcal{X}\mu_{k+1} + P_\mathcal{X}\mu_{k+1} - P_\mathcal{X}\mu_k\|_1 \\
&\leq \|\mathcal{I}_\mathcal{X}(\mu_k - \mu_{k+1})\|_1 + \|P_\mathcal{X}(\mu_{k+1} - \mu_k)\|_1 \\
&\leq 2\|\mu_k - \mu_{k+1}\|_1,
\end{aligned}
$$

where we have used the fact that both $\mathcal{I}_\mathcal{X}$ and $P_\mathcal{X}$ are non-expansions for the $\ell_1$-norm in the last line, which follows from the data-processing inequality for the total variation distance. We then conclude the analysis for the even rounds by replacing $\mu_{k+1}$ with $\mu_{k-1}$ in the argument above, and repeating the same reasoning for odd rounds completes the overall proof. $\qquad\square$

Having established these elementary results, we now turn to addressing the main technical hurdle: bounding the cumulative rounding errors.

**Theorem 4.** *The cumulative constraint violations of the iterates produced by Sinkhorn Value Iteration satisfy*

$$\sum_{k=1}^{K} \delta(\mu_k) \leq \frac{(1-\gamma)\mathcal{H}\left(\mu^* \| \mu_1\right)}{2\eta \left\| c \right\|_\infty} + \frac{16\eta \left\| c \right\|_\infty K}{(1-\gamma)^2}.$$

*Proof.* We start by applying Lemma 6 to show $\delta(\mu_k) \leq 2 \left\| \mu_k - \mu_{k+1} \right\|_1$, which reduces our task to bounding $\sum_{k=1}^{K} \left\| \mu_k - \mu_{k+1} \right\|_1$. We do this as follows, for any fixed $\alpha > 0$:

$$\sum_{k=1}^{K} \left\| \mu_k - \mu_{k+1} \right\|_1 \leq \frac{\alpha K}{2} + \sum_{k=1}^{K} \frac{\left\| \mu_k - \mu_{k+1} \right\|_1^2}{2\alpha}$$

$$\text{(by the inequality of arithmetic and geometric means)}$$

$$\leq \frac{\alpha K}{2} + \sum_{k=1}^{K} \frac{\mathcal{D}\left(\mu_{k+1} \| \mu_k\right)}{\alpha}$$

$$\text{(Pinsker's inequality)}$$

$$\leq \frac{\alpha K}{2} + \sum_{k=1}^{K} \frac{\mathcal{H}\left(\mu_{k+1} \| \mu_k\right)}{(1-\gamma)\alpha}$$

$$\text{(Lemma 3)}$$

$$\leq \frac{\alpha K}{2} + \sum_{k=1}^{K} \frac{\mathcal{H}\left(\mu^* \| \mu_k\right) - \mathcal{H}\left(\mu^* \| \mu_{k+1}\right)}{(1-\gamma)\alpha} + \frac{\eta}{\alpha\left(1-\gamma\right)} \sum_{k=1}^{K} \left\langle c, \mu^* - \mu_{k+1} \right\rangle$$

$$\text{(Lemma 2)}$$

$$\leq \frac{\alpha K}{2} + \frac{\mathcal{H}\left(\mu^* \| \mu_1\right) - \mathcal{H}\left(\mu^* \| \mu_{K+1}\right)}{\alpha(1-\gamma)} + \frac{\eta}{\alpha\left(1-\gamma\right)} \sum_{k=1}^{K} \left\langle c, \rho(\mu_{k+1}) - \mu_{k+1} \right\rangle$$

$$\text{(since } \langle \mu^*, c \rangle \leq \langle \rho(\mu_{k+1}), c \rangle)$$

$$\leq \frac{\alpha K}{2} + \frac{\mathcal{H}\left(\mu^* \| \mu_1\right)}{\alpha(1-\gamma)} + \frac{\eta \left\| c \right\|_\infty}{\alpha(1-\gamma)^2} \sum_{k=1}^{K} \Delta(\mu_{k+1})$$

$$\text{(Lemma 4)}$$

$$\leq \frac{\alpha K}{2} + \frac{\mathcal{H}\left(\mu^* \| \mu_1\right)}{\alpha(1-\gamma)} + \frac{2\eta \left\| c \right\|_\infty}{\alpha(1-\gamma)^2} \sum_{k=1}^{K} \delta(\mu_{k+1})$$

$$\text{(Lemma 5)}$$

$$\leq \frac{\alpha K}{2} + \frac{\mathcal{H}\left(\mu^* \| \mu_1\right)}{\alpha(1-\gamma)} + \frac{4\eta \left\| c \right\|_\infty}{\alpha(1-\gamma)^2} \sum_{k=1}^{K} \left\| \mu_k - \mu_{k+1} \right\|_1,$$

where we have finally used that $\delta(\mu_{k+1}) \leq 2 \left\| \mu_k - \mu_{k+1} \right\|_1$ (Lemma 6). Now, we need to make sure that $\frac{4\eta\|c\|_\infty}{\alpha(1-\gamma)^2} \leq 1$ to turn this into a meaningful result. In particular, setting $\alpha = \frac{8\eta\|c\|_\infty}{(1-\gamma)^2}$ guarantees that the constant in question equals $\frac{1}{2}$, so that we can reorder the terms to obtain

$$\sum_{k=1}^{K} \left\| \mu_k - \mu_{k+1} \right\|_1 \leq \alpha K + \frac{2\mathcal{H}\left(\mu^* \| \mu_1\right)}{\alpha(1-\gamma)} = \frac{8\eta \left\| c \right\|_\infty K}{(1-\gamma)^2} + \frac{(1-\gamma)\mathcal{H}\left(\mu^* \| \mu_1\right)}{4\eta \left\| c \right\|_\infty}.$$

This concludes the proof. □

### C.3   Regret analysis

In this section, we bound the regret of the iterates produced by Sinkhorn Value Iteration.

**Theorem 5.** *The regret of the mirror Sinkhorn procedure satisfies*

$$\sum_{k=1}^{K} \langle \mu_k - \mu^*, c \rangle \leq \frac{\mathcal{H}\left(\mu^* \| \mu_1\right)}{\eta} + 2 \left\| c \right\|_{\infty}.$$

*Proof.* We first apply Lemma 2 to obtain the bound

$$\langle \mu_k - \mu^*, c \rangle \leq \frac{\mathcal{H}\left(\mu \| \mu_k\right) - \mathcal{H}\left(\mu \| \mu_{k+1}\right) - \mathcal{H}\left(\mu_{k+1} \| \mu_k\right)}{\eta} + \langle c, \mu_{k+1} - \mu_k \rangle. \qquad (13)$$

Adding up both sides for all $k = 1, 2, \ldots, K$, we get

$$\sum_{k=1}^{K} \langle \mu_k - \mu^*, c \rangle \leq \frac{\mathcal{H}\left(\mu \| \mu_1\right) - \mathcal{H}\left(\mu \| \mu_{K+1}\right) - \sum_{k=1}^{K} \mathcal{H}\left(\mu_{k+1} \| \mu_k\right)}{\eta} + \langle c, \mu_{K+1} - \mu_1 \rangle$$

$$\leq \frac{\mathcal{H}\left(\mu \| \mu_1\right)}{\eta} + 2 \left\| c \right\|_{\infty}.$$

This concludes the proof. $\qquad \square$

### C.4 The proof of Theorem 2

The proof follows from applying the above results to bounding the rounding errors and the regret. The first of these is handled as follows:

$$\langle \mu_{\text{out}} - \overline{\mu}_K, c \rangle \leq \left\| c \right\|_{\infty} \frac{\Delta(\overline{\mu}_K)}{1 - \gamma} \leq 2 \left\| c \right\|_{\infty} \frac{\delta(\overline{\mu}_K)}{1 - \gamma}$$

$$\leq 2 \left\| c \right\|_{\infty} \frac{\sum_{k=1}^{K} \delta(\mu_k)}{K(1 - \gamma)} \leq \frac{\mathcal{H}\left(\mu^* \| \mu_1\right)}{K\eta} + \frac{32\eta \left\| c \right\|_{\infty}^2}{(1 - \gamma)^3},$$

where the first and second inequalities respectively come from Lemmas 4 and 5, the third one from the convexity of $\delta$, and the last inequality comes from the application of Theorem 4.

The regret is then bounded using Theorem 5 as

$$\langle \overline{\mu}_K - \mu^*, c \rangle = \frac{1}{K} \sum_{k=1}^{K} \langle \mu_k - \mu^*, c \rangle \leq \frac{\mathcal{H}\left(\mu^* \| \mu_1\right)}{K\eta} + \frac{2}{K} \left\| c \right\|_{\infty}.$$

Putting both bounds together, we obtain

$$\langle \mu_{\text{out}} - \mu^*, c \rangle = \langle \mu_{\text{out}} - \overline{\mu}_K, c \rangle + \langle \overline{\mu}_K - \mu^*, c \rangle$$

$$\leq \frac{2\mathcal{H}\left(\mu^* \| \mu_1\right)}{K\eta} + \frac{32\eta \left\| c \right\|_{\infty}^2}{(1 - \gamma)^3} + \frac{2 \left\| c \right\|_{\infty}}{K}$$

Now, to prove the actual claim of the theorem, we now let $\pi_1$ be the uniform transition coupling and $\mu_1$ be the associated occupancy coupling. In this case, the conditional relative entropy can be upper bounded as $\mathcal{H}\left(\mu^* \| \mu_1\right) \leq \log |\mathcal{X}||\mathcal{Y}|$, and we can further bound

$$\langle \mu_{\text{out}} - \mu^*, c \rangle \leq \frac{2 \log |\mathcal{X}||\mathcal{Y}|}{K\eta} + \frac{32\eta \left\| c \right\|_{\infty}^2}{(1 - \gamma)^3} + \frac{2 \left\| c \right\|_{\infty}}{K}$$

$$\leq 16 \left\| c \right\|_{\infty} \sqrt{\frac{\log |\mathcal{X}||\mathcal{Y}|}{K(1 - \gamma)^3}} + \frac{2 \left\| c \right\|_{\infty}}{K}$$

$$\leq 18 \left\| c \right\|_{\infty} \sqrt{\frac{\log |\mathcal{X}||\mathcal{Y}|}{K(1 - \gamma)^3}},$$

where the second to last line is obtained by picking $\eta = \frac{1}{4\|c\|_\infty}\sqrt{\frac{(1-\gamma)^3 \log|\mathcal{X}||\mathcal{Y}|}{K}}$ and the last line is obtained by noticing that $\frac{1}{K} \le \sqrt{\frac{\log|\mathcal{X}||\mathcal{Y}|}{K(1-\gamma)^3}}$ holds whenever $K \ge 1$, $|\mathcal{X}| \ge 2$, $|\mathcal{Y}| \ge 2$ and $\gamma \in (0,1)$. Finally, note that

$$V^{\pi_{\text{out}}}(x_0 y_0) - \mathbb{W}_\gamma(M_\mathcal{X}, M_\mathcal{Y}; c, x_0, y_0) = \frac{1}{1-\gamma}\langle \mu_{\text{out}} - \mu^*, c \rangle \le 18\|c\|_\infty \sqrt{\frac{\log|\mathcal{X}||\mathcal{Y}|}{K(1-\gamma)^5}}.$$

Now, it can be directly verified that the right-hand side is indeed at most $\varepsilon$ whenever $K$ is greater than the expression given in the theorem, thus concluding the proof. □

# D  Sinkhorn Policy Iteration

We describe here a simple alternative to Sinkhorn Value Iteration called Sinkhorn Policy Iteration (SPI). After introducing this method heuristically, we provide a formal performance analysis, and finally explain its relation to SVI.

The core concept underlying the definition of SPI is the notion of *Q-functions*, defined analogously to action-value functions in an MDP. The Q-function associated with a transition coupling $\pi$ is a function $Q^\pi : \mathcal{X}\mathcal{Y} \times \mathcal{X}\mathcal{Y} \to \mathbb{R}$, with each of its entries defined as

$$Q^\pi(xy, x'y') = \mathbb{E}\left[\sum_{t=0}^{\infty} \gamma^t c(X_t, Y_t) \,\middle|\, (X_0, Y_0) = (x, y), (X_1, Y_1) = (x'y')\right]$$

Analogously to the results presented in Section B.1, it is possible to show that the Q-function of a given transition coupling $\pi$ satisfies the Bellman equations

$$Q^\pi(xy, x'y') = c(xy) + \gamma \sum_{x''y''} \pi(x''y''|x'y')Q^\pi(x'y', x'', y''),$$

and that the Q-value function of the optimal transition coupling $Q^* = Q^{\pi^*}$ satisfies the Bellman optimality equations

$$Q^*(xy, x'y') = c(xy) + \gamma \inf_{p \in \Pi_{x'y'}} \sum_{x''y''} p(x''y'')Q^*(x'y', x'', y'').$$

These are respectively the fixed points of the Bellman operator $\mathcal{T}^\pi : \mathbb{R}^{\mathcal{X}\mathcal{Y} \times \mathcal{X}\mathcal{Y}} \to \mathbb{R}^{\mathcal{X}\mathcal{Y} \times \mathcal{X}\mathcal{Y}}$ defined via

$$(\mathcal{T}^\pi f)(xy, x'y') = c(xy) + \gamma \sum_{x''y''} \pi(x''y''|x'y')f(x'y', x'', y'')$$

and the Bellman optimality operator $\mathcal{T} : \mathbb{R}^{\mathcal{X}\mathcal{Y} \times \mathcal{X}\mathcal{Y}} \to \mathbb{R}^{\mathcal{X}\mathcal{Y} \times \mathcal{X}\mathcal{Y}}$ defined via

$$(\mathcal{T}f)(xy, x'y') = c(xy) + \gamma \inf_{p \in \Pi_{x'y'}} \sum_{x''y''} p(x''y'')f(x'y', x'', y'').$$

The system of equations $\mathcal{T}Q^* = Q^*$ is essentially as hard to solve (or even harder) than the Bellman optimality equations for $V^*$ stated earlier. However, one can develop an algorithmic approach toward finding an optimal transition coupling by drawing inspiration from the literature on regularized dynamic programming.

In particular, we develop below an analogue of an entropy-regularized policy iteration scheme that is known under many names in the RL literature: Natural Policy Gradients by Kakade [2001] and Agarwal et al. [2021a], MDP-Expert by Even-Dar et al. [2009], Mirror-Descent Policy Iteration by Geist et al. [2019], POLITEX by Abbasi-Yadkori et al. [2019], Policy Mirror Descent by Agarwal et al. [2021a], and the list goes on. This method can be directly adapted to our setting as follows. Starting from an arbitrary initial transition coupling $\pi_1$, SPI performs the following sequence of updates for each $k = 1, 2, \ldots, K$:

- Round the transition coupling $\pi_k$ to $\widetilde{\pi}_k = \rho(\pi_k)$,
- update the Q-function by solving the fixed-point equation $Q_k = \mathcal{T}^{\widetilde{\pi}_k} Q_k$,
- if $k$ is odd, then update the transition coupling as

$$\pi_{k+1}(x'y'|xy) = \frac{\pi_k(x'y'|xy)\exp(-\eta Q_k(xy, x'y'))}{\sum_{y''} \pi_k(x'y''|xy)\exp(-\eta Q_k(xy, x'y''))} P_\mathcal{X}(x'|x),$$

- else if $k$ is even, then update the transition coupling as

$$\pi_{k+1}(x'y'|xy) = \frac{\pi_k(x'y'|xy)\exp(-\eta Q_k(xy, x'y'))}{\sum_{x''} \pi_k(x''y'|xy)\exp(-\eta Q_k(xy, x''y'))} P_\mathcal{Y}(y'|y).$$

As in the case of SVI, it is easy to verify that these transition couplings satisfy the required marginal constraints. Furthermore, one can verify that the updates defined above exactly correspond to running an instance of Mirror Sinkhorn [Ballu and Berthet, 2023] in each state-pair $xy$ with the sequence of cost functions $Q_1(xy, \cdot), Q_2(xy, \cdot), \ldots, Q_K(xy, \cdot)$, similarly how the entropy-regularized policy iteration methods run an instance of entropic mirror descent in each state. We discuss various aspects of this algorithm below.

## D.1 Practical implementation

Like SVI, this method can be seen as performing online Mirror Sinkhorn updates in each state pair $xy$ with a sequence of cost functions $Q_k$, which are computed via solving the linear system of Bellman equations $Q_k = \mathcal{T}^{\widetilde{\pi}_k} Q_k$. The occupancy couplings produced by SPI are denoted by $\widetilde{\mu}_k = \mu^{\widetilde{\pi}_k}$, and the final output of the method is produced by averaging these occupancies as $\mu_{\text{out}} = \frac{1}{K} \sum_{k=1}^{K} \widetilde{\mu}_k$, and then extracting the transition coupling $\pi_{\text{out}} = \pi_{\mu_{\text{out}}}$. Notably, there is no need to round this transition coupling since each $\widetilde{\mu}_k$ satisfies all constraints by construction, and so does their average $\mu_{\text{out}}$ due to the linearity of the constraints.

---
**Algorithm 2:** Sinkhorn Policy Iteration

**Input:** $P_{\mathcal{X}}, P_{\mathcal{Y}}, c, \eta, \gamma, K, m$
**Initialise:** $\pi_1 \leftarrow P_{\mathcal{X}} \otimes P_{\mathcal{Y}}$;
**for** $k = 1, ..., K-1$ **do**
  $\quad \widetilde{\pi}_k \leftarrow \rho(\pi_k)$;
  $\quad Q \leftarrow (\mathcal{T}^{\widetilde{\pi}_k})^m Q$;
  $\quad \pi_{k+1} \leftarrow \textbf{update}(\pi_k, Q)$;  {Equation 10}
**end**
$\mu_{\text{out}} \leftarrow \frac{1}{K} \sum_{k=1}^{K} \widetilde{\mu}_k$;
$\pi_{\text{out}} \leftarrow \pi_{\mu_{\text{out}}}$;
$V^{\pi_{\text{out}}} \leftarrow \textbf{evaluate}(\pi_{\text{out}})$;
**Output:** $\pi_{\text{out}}, V^{\pi_{\text{out}}}$  {Final coupling}

---

One advantage of SPI over SVI is that finding an exact solution for the fixed-point equation $Q_k = \mathcal{T}^{\pi_k} Q_k$ is easier than computing the fixed points required by SVI, thanks to the fact that this is a linear system of equations. A downside of the method is that it requires to run the rounding procedure after each update, at least for the theoretical guarantees to remain valid. The impact of these steps may however be negligible in practical implementations. Similarly to SVI, the ideal updates of SPI can be approximated by applying the Bellman operator to the Q-functions only a small number of times $m$, with $m = \infty$ corresponding to the ideal implementation analyzed above. We present a pseudocode for SPI as Algorithm 2.

## D.2 Convergence guarantees

In what follows, we show the following performance guarantee for SPI.

**Theorem 6.** *Suppose that Sinkhorn Policy Iteration is run for $K$ steps with regularization parameter* $\eta = \frac{1-\gamma}{3\|c\|_\infty} \sqrt{\frac{8 \log |\mathcal{X}||\mathcal{Y}|}{K}}$, *and initialized with the uniform coupling defined for each* $xy, x'y'$ *as* $\pi_1(x'y'|xy) = \frac{1}{|\mathcal{X}||\mathcal{Y}|}$. *Then, for any* $x_0 y_0 \in \mathcal{X}\mathcal{Y}$, *the output satisfies* $V^{\pi_{out}}(x_0 y_0) \leq \mathbb{W}_\gamma(M_{\mathcal{X}}, M_{\mathcal{Y}}; c, x_0, y_0) + \varepsilon$ *if the number of iterations is at least*

$$K \geq \frac{5 \|c\|_\infty^2 \log |\mathcal{X}||\mathcal{Y}|}{(1-\gamma)^4 \varepsilon^2}.$$

As the analyses of all regularized policy iteration methods listed above, this one also starts with establishing the following claim that corresponds to the classic *performance difference lemma* (often attributed to Kakade and Langford, 2002, but proposed much earlier in works like Cao, 1999 and even Howard, 1960). To state the result, we define $V_k(xy) = \sum_{x'y'} \widetilde{\pi}_k(x'y'|xy) Q_k(xy, x'y')$ and the operators $E_+ : \mathbb{R}^{\mathcal{X}\mathcal{Y}\times\mathcal{X}\mathcal{Y}} \to \mathbb{R}^{\mathcal{X}\mathcal{Y}}$ and $E_- : \mathbb{R}^{\mathcal{X}\mathcal{Y}\times\mathcal{X}\mathcal{Y}} \to \mathbb{R}^{\mathcal{X}\mathcal{Y}}$ with each element given by $(E_- V)(xy, x'y') = V(xy)$ and $(E_+ V)(xy, x'y') = V(x'y')$. Then, the following bound holds on the instantaneous regret of SPI in round $k$.

**Lemma 7.** $\langle \widetilde{\mu}_k - \mu^*, c \rangle = \langle \mu^*, E_- V_k - Q_k \rangle$.

*Proof.* The proof follows from elementary properties of occupancy couplings. First, note that by the definition of the value function $V_k$, we have

$$\langle \widetilde{\mu}_k, c \rangle = (1-\gamma) \langle \nu_0, V_k \rangle.$$

Furthermore, by multiplying both sides of the Bellman equations $Q_k = c + \gamma E_+ V_k$ with $\mu^*$ and using the flow constraints $E_+^{\mathsf{T}} \mu^* = \gamma E_-^{\mathsf{T}} \mu^* + (1-\gamma)\nu_0$, we obtain

$$\langle \mu^*, Q_k \rangle = \langle \mu^*, c + \gamma E_+ V_k \rangle = \langle \mu^*, c + E_- V_k \rangle - (1-\gamma) \langle \nu_0, V_k \rangle.$$

The result follows after reordering the terms. $\square$

Given the above lemma, we can readily express the regret of SPI as follows:

$$\sum_{k=1}^{K} \langle \widetilde{\mu}_k - \mu^*, c \rangle = \sum_{k=1}^{K} \langle \mu^*, E_- V_k - Q_k \rangle$$

$$= \sum_{xy} \nu^*(xy) \sum_{k=1}^{K} \langle \pi^*(\cdot|xy) - \widetilde{\pi}_k(\cdot|xy), Q_k(xy, \cdot) \rangle,$$

where we can recognize the regret of Mirror Sinkhorn in each state pair $xy \in \mathcal{XY}$. Thus, applying the bound of Theorem 3.1 of Ballu and Berthet [2023] to each of these terms (while noting that $\|Q_k\|_\infty \le \|c\|_\infty / (1 - \gamma)$ holds for all $k$) gives

$$\sum_{k=1}^{K} \langle \pi^*(\cdot|xy) - \widetilde{\pi}_k(\cdot|xy), Q_k(xy, \cdot) \rangle \le \frac{\mathcal{D}_{\mathrm{KL}}\left(\pi^*(\cdot|xy)\|\pi_1(\cdot|xy)\right)}{\eta} + \frac{9\eta \|c\|_\infty^2 K}{8(1 - \gamma)^2},$$

and thus putting the bounds together we obtain

$$\langle \mu_{\mathrm{out}} - \mu^*, c \rangle = \frac{1}{K} \sum_{k=1}^{K} \langle \widetilde{\mu}_k - \mu^*, c \rangle \le \frac{\mathcal{H}\left(\mu^*\|\mu_1\right)}{\eta K} + \frac{9\eta \|c\|_\infty^2}{8(1 - \gamma)^2}$$

Now, setting $\pi_1$ as the uniform coupling, we can further upper bound the conditional relative entropy as $\mathcal{H}\left(\mu^*\|\mu_1\right) \le \log |\mathcal{X}||\mathcal{Y}|$, and after setting $\eta = \frac{1-\gamma}{3\|c\|_\infty}\sqrt{\frac{8 \log |\mathcal{X}||\mathcal{Y}|}{K}}$, the bound becomes

$$\langle \mu_{\mathrm{out}} - \mu^*, c \rangle \le \frac{6 \|c\|_\infty}{1 - \gamma} \sqrt{\frac{\log |\mathcal{X}||\mathcal{Y}|}{8K}}.$$

Finally, note that

$$V^{\pi_{\mathrm{out}}}(x_0 y_0) - \mathbb{W}_\gamma(M_\mathcal{X}, M_\mathcal{Y}; c, x_0, y_0) = \frac{1}{1 - \gamma} \langle \mu_{\mathrm{out}} - \mu^*, c \rangle \le \frac{6 \|c\|_\infty}{(1 - \gamma)^2} \sqrt{\frac{\log |\mathcal{X}||\mathcal{Y}|}{8K}}.$$

Using a crude upper bound $36/8 \le 5$ verifies the claim of Theorem 6.

### D.3 Relation to Sinkhorn Value Iteration

While on the surface, SPI may seem only loosely related to SVI, a closer connection can be drawn by making the following observations. First, observe that the transition-coupling updates exactly match the updates of Sinkhorn Value Iteration, although there is an apparent difference in how the Q-functions are defined. To expose the similarity between the two methods better, let us consider an even round in which $\pi_k$ satisfies the $\mathcal{X}$-marginal conditions $\sum_{y'} \pi_k(x'y'|xy) = P(x'|x)$. Thus, introducing the notation $V_\mathcal{X}(xy, x') = \sum_{y'} \frac{\pi_k(x'y'|xy)}{P(x'|x)} Q_k(xy, x'y')$, we can multiply the Bellman equations by $\pi_k(x'y'|xy)/P(x'|x)$ and sum them up to obtain

$$V_\mathcal{X}(xy, x') = \sum_{y'} \frac{\pi_k(x'y'|xy)}{P(x'|x)} \left( c(xy) + \gamma \sum_{x''y''} \pi_k(x''y''|x'y')Q_k(x'y', x''y'') \right)$$

$$= \sum_{y'} \frac{\pi_k(x'y'|xy)}{P(x'|x)} \left( c(xy) + \gamma \sum_{x''} P(x''|x')V_\mathcal{X}(x'y', x'') \right)$$

$$\approx -\frac{1}{\eta} \log \sum_{y'} \frac{\pi_k(x'y'|xy)}{P(x'|x)} \exp\left( -\eta \left( c(xy) + \gamma \sum_{x''} P(x''|x')V_\mathcal{X}(x'y', x'') \right) \right),$$

where the approximation is accurate as $\eta$ approaches zero. Thus, for small values of $\eta$, the Sinkhorn–Bellman operator used by Sinkhorn Value Iteration is an accurate approximation of the Bellman operator used by Sinkhorn Policy Iteration, and thus one may reasonably expect their respective fixed points to be close as well (this intuition may, however easily fail and is not necessary for our analysis above).

# E   Auxiliary proofs and technical results

In this appendix we prove several technical results from the main text.

## E.1   Proof of Proposition 1

We prove the claim by showing that the solution of the constrained optimization problem of Equation (9) is equivalent to the transition-coupling update rule specified in Equation (10). To this end, we study the Lagrangian of the optimization problem (9) corresponding to the update for odd rounds, $\mathcal{B}_\mathcal{X}$, and note that the update rule for even rounds, $\mathcal{B}_\mathcal{Y}$, can be worked out analogously. By introducing Lagrange multipliers $V_\mathcal{X}(xy, x')$ for each constraint in $\mathcal{B}_\mathcal{X}$, we obtain the Lagrangian

$$\mathcal{L}(\mu; V_\mathcal{X}) = \langle \mu, c \rangle + \frac{1}{\eta} \mathcal{H}\left(\mu \| \mu_k\right)$$

$$+ \sum_{xy,x'} V_\mathcal{X}(xy, x') \left( \left( \gamma \sum_{x''y''} \mu(x''y'', xy) + (1-\gamma)\nu_0(xy) \right) P_\mathcal{X}(x'|x) - \sum_{y'} \mu(xy, x'y') \right)$$

$$= \sum_{xy,x'y'} \mu(xy, x'y') \left( c(xy) + \gamma \sum_{x''} P_\mathcal{X}(x''|x') V_\mathcal{X}(x'y', x'') - V_\mathcal{X}(xy, x') \right)$$

$$+ (1-\gamma) \sum_{xy,x'} \nu_0(xy) P_\mathcal{X}(x'|x) V_\mathcal{X}(xy, x') + \frac{1}{\eta} \mathcal{H}\left(\mu \| \mu_k\right).$$

A quick calculation (cf. Appendix A.1 of Neu et al., 2017) shows that the derivative of $\mathcal{H}\left(\mu \| \mu_k\right)$ satisfies

$$\frac{\partial \mathcal{H}\left(\mu \| \mu_k\right)}{\partial \mu(xy, x'y')} = \log \pi_\mu(x'y'|xy) - \log \pi_{\mu_k}(x'y'|xy) = \log \pi_\mu(x'y'|xy) - \log \pi_k(x'y'|xy),$$

where we have used that $\pi_{\mu_k} = \pi_k$ holds by definition of $\mu_k$ and $\pi_k$. To proceed, for a fixed $V_\mathcal{X}$, we set the gradient of the Lagrangian to zero and solve for the transition coupling $\pi_{k+1}$, which gives

$$\pi_{k+1}(x'y'|xy) = \pi_k(x'y'|xy) \exp\left( -\eta \left( c(xy) + \gamma \sum_{x''} P_\mathcal{X}(x''|x') V_\mathcal{X}(x'y', x'') - V_\mathcal{X}(xy, x') \right) \right).$$

Then, the correct choice of $V_\mathcal{X}$ has to be such that the constraint $\sum_{y'} \pi_{k+1}(x'y'|xy) = P_\mathcal{X}(x'|x)$ is satisfied. To see this, suppose that this condition is indeed verified and that $\mu_{k+1}$ is the occupancy coupling associated with $\pi_{k+1}$. We need to show that $\mu_{k+1}$ indeed verifies the condition defining $\mathcal{B}_\mathcal{X}$. To this end, notice that $\mu_{k+1}$ satisfies Equation (6), and thus the condition can be simply written as

$$\sum_{y'} \mu_{k+1}(xy, x'y') = \left( \sum_{x''y''} \mu_{k+1}(xy, x''y'') \right) P_\mathcal{X}(x'|x),$$

which, after recalling the relation $\pi_{k+1}(x'y'|xy) = \frac{\mu(xy,x'y')}{\sum_{x''y''} \mu(xy,x''y'')}$ can be indeed seen to hold if $\sum_{y'} \pi_{k+1}(x'y'|xy) = P_\mathcal{X}(x'|x)$ is true.

Enforcing this constraint gives the following expression for $V_\mathcal{X}(xy, x')$:

$$V_\mathcal{X}(xy, x') = -\frac{1}{\eta} \log \sum_{y'} \frac{\pi_k(x'y'|xy)}{P_\mathcal{X}(x'|x)} \exp\left( -\eta \left( c(xy) + \gamma \sum_{x''} P_\mathcal{X}(x''|x') V_\mathcal{X}(x'y', x'') \right) \right).$$

To conclude, we need to ensure that this system of equations has a unique solution. In order to do this, we recall the definition of the Bellman–Sinkhorn operator $\mathcal{T}_\mathcal{X}^{\pi_k} : \mathbb{R}^{\mathcal{X}\mathcal{Y}\times\mathcal{X}} \to \mathbb{R}^{\mathcal{X}\mathcal{Y}\times\mathcal{X}}$, acting on a function $f$ as

$$(\mathcal{T}_\mathcal{X}^{\pi_k} f)(x'y', x'') = -\frac{1}{\eta} \log \sum_{y'} \frac{\pi_k(x'y'|xy)}{P_\mathcal{X}(x'|x)} \exp\left( -\eta \left( c(xy) + \gamma \sum_{x''} P_\mathcal{X}(x''|x') f(x'y', x'') \right) \right).$$

With this notation, we can directly verify that $V_{\mathcal{X}}$ satisfies $\mathcal{T}_{\mathcal{X}}^{\pi_k} V_{\mathcal{X}} = V_{\mathcal{X}}$. Furthermore, it can be shown that the operator $\mathcal{T}_{\mathcal{X}}^{\pi_k}$ is a $\gamma$-contraction in supremum norm (cf. Lemma 8), and thus it has a unique fixed point by the Banach fixed-point theorem. We finally note that defining $Q_k(xy, x'y') = (c(xy) + \gamma \sum_{x''} P_{\mathcal{X}}(x''|x')V_{\mathcal{X}}(x'y', x''))$, the transition-coupling update derived above can be rewritten as

$$\pi_{k+1}(x'y'|xy) = \frac{\pi_k(x'y'|xy) \exp\left(-\eta Q_k(xy, x'y')\right)}{\sum_{y'} \pi_k(x'y''|xy) \exp\left(-\eta Q_k(xy, x'y'')\right)} P_{\mathcal{X}}(x'|x). \tag{14}$$

This concludes the proof. □

### E.2 Rounding procedure and proof of Lemma 5

We adapt the rounding procedure stated as Algorithm 2 of Altschuler et al. [2017] and reproduced here as Algorithm 3 (where / denotes element-wise division in the pseudocode). Formally, for two probability distributions $p \in \Delta(\mathcal{X})$ and $q \in \Delta(\mathcal{Y})$, the set of valid couplings is $\mathcal{U}_{p,q} = \{P \in \mathbb{R}_+^{\mathcal{X}\mathcal{Y}} : P \cdot \mathbf{1} = p;$ $P^T \cdot \mathbf{1} = q\}$. For a nonnegative matrix $F \in \mathbb{R}_+^{\mathcal{X}\mathcal{Y}}$, the rounding procedure outputs a valid coupling $\rho(F, p, q) \in \mathcal{U}_{p,q}$ which, by Lemma 7 of Altschuler et al. [2017], satisfies

---
**Algorithm 3:** Rounding procedure for couplings

**Input:** approximate coupling $F$, margins $p$, $q$
$X \leftarrow \mathrm{diag}(\min(p/(F \cdot \mathbf{1}), \mathbf{1}))$;
$F' \leftarrow XF$;
$Y \leftarrow \mathrm{diag}(\min(q/(F'^{\top} \cdot \mathbf{1}), \mathbf{1}))$;
$F'' \leftarrow F'Y$;
$\mathrm{err}_p = p - F'' \cdot \mathbf{1},\ \mathrm{err}_q = q - F''^{\top} \cdot \mathbf{1}$;
**Output:** $G \leftarrow F'' + \mathrm{err}_p \mathrm{err}_q^{\top} / \|\mathrm{err}_p\|_1$

---

$$\|\rho(F, p, q) - F\|_1 \leq 2\left(\|F \cdot \mathbf{1} - p\|_1 + \|F^T \cdot \mathbf{1} - q\|_1\right).$$

We will now define the rounding procedure for a (not necessarily valid) transition coupling $\pi \in \mathbb{R}^{\mathcal{X}\mathcal{Y} \times \mathcal{X}\mathcal{Y}}$ by using the aforementioned rounding procedure at each state pair as

$$\tilde{\pi}(\cdot|xy) = \rho(\pi(\cdot|xy), P_{\mathcal{X}}(\cdot|x), P_{\mathcal{Y}}(\cdot|y)).$$

With some abuse of notation, we will write the resulting transition coupling as $\widetilde{\pi} = \rho(\pi)$ and the associated occupancy coupling as $\widetilde{\mu} = \mu^{\widetilde{\pi}} = \rho(\mu)$. Because of the correctness of the original rounding procedure, this transition coupling is valid, and so is the associated occupancy coupling. We can now proceed to the proof of Lemma 5.

*Proof of Lemma 5.* Let $\mu \in \mathbb{R}_+^{\mathcal{X}\mathcal{Y} \times \mathcal{X}\mathcal{Y}}$, and as before define $\nu_\mu(xy) = \sum_{x'y'} \mu(xy, x'y')$ and

$$\pi_\mu(x'y'|xy) = \begin{cases} \frac{\mu(xy, x'y')}{\nu_\mu(xy)} & \text{if } \nu_\mu(xy) \neq 0, \\ P_{\mathcal{X}}(x'|x)P_{\mathcal{Y}}(y'|y) & \text{otherwise.} \end{cases}$$

For arbitrary state pairs $xy$, we use Lemma 7 of Altschuler et al. [2017] to obtain that

$$\|\tilde{\pi}_\mu(\cdot|xy) - \pi_\mu(\cdot|xy)\|_1 \leq 2\left[\sum_{x'}\left|P_{\mathcal{X}}(x'|x) - \sum_{y'}\pi_\mu(x'y'|xy)\right| + \sum_{y'}\left|P_{\mathcal{Y}}(y'|y) - \sum_{x'}\pi_\mu(x'y'|xy)\right|\right].$$

Now, multiplying by $\nu_\mu(xy)$ and summing over $xy$, we get

$$\sum_{xy} \nu_\mu(xy) \|\tilde{\pi}_\mu(\cdot|xy) - \pi_\mu(\cdot|xy)\|_1$$

$$\leq 2\left[\sum_{xyx'}\nu_\mu(xy)\left|P_{\mathcal{X}}(x'|x) - \sum_{y'}\pi_\mu(x'y'|xy)\right| + \sum_{xyy'}\nu_\mu(xy)\left|P_{\mathcal{Y}}(y'|y) - \sum_{x'}\pi_\mu(x'y'|xy)\right|\right]$$

$$= 2\left[\sum_{xyx'}\left|\nu_\mu(xy)P_{\mathcal{X}}(x'|x) - \sum_{y'}\mu(xy, x'y')\right| + \sum_{xyy'}\left|\nu_\mu(xy)P_{\mathcal{Y}}(y'|y) - \sum_{x'}\mu(xy, x'y')\right|\right]$$

$$= 2\delta_{\mathcal{X}}(\mu) + 2\delta_{\mathcal{Y}}(\mu) = 2\delta(\mu),$$

where we used the fact that $\mu(xy, x'y') = \nu_\mu(xy)\pi_\mu(x'y'|xy)$ for any $xy, x'y'$, and the definitions $\delta_{\mathcal{X}}$, $\delta_{\mathcal{Y}}$ and $\delta$. Finally, we notice that $\Delta(\mu) = \sum_{xy} \nu_\mu(xy) \|\tilde{\pi}_\mu(\cdot|xy) - \pi_\mu(\cdot|xy)\|_1$, which concludes the proof. □

### E.3 The contraction property of the Bellman–Sinkhorn operator

**Lemma 8.** *Let $\pi : \mathcal{XY} \to \Delta_{\mathcal{XY}}$ be arbitrary and consider the associated Bellman–Sinkhorn operator $\mathcal{T}^\pi$ acting on a function $f : \mathcal{XY} \times \mathcal{Y} \to \mathbb{R}$ as*

$$(\mathcal{T}^\pi f)(x'y', x'') = -\frac{1}{\eta} \log \sum_{y'} \frac{\pi(x'y'|xy)}{P(x'|x)} \exp\left(-\eta\left(c(xy) + \gamma \sum_{x''} P(x''|x') f(x'y', x'')\right)\right).$$

*Then, $\mathcal{T}^\pi$ is a $\gamma$-contraction for the supremum norm $\|\cdot\|_\infty$, that is, for any two functions $f_1, f_2 : \mathcal{XY} \times \mathcal{X}$, we have*

$$\|\mathcal{T}^\pi f_1 - \mathcal{T}^\pi(f_2)\|_\infty \leq \gamma \|f_1 - f_2\|_\infty.$$

*Proof.* The claim easily follows from using the standard fact that the function $g_p(z) = \log \sum_{y'} p(y') e^z(y')$ is 1-smooth with respect to the supremum norm, so that for any two vectors $z$, we have $|g_p(z) - g_p(z)| \leq \|z - z'\|_\infty$. To apply this result, we define $q(y'|xy) = \pi(x'y'|xy)/P(x'|x)$ and $z_1(xy, x'y') = c(xy) + \gamma \sum_{x''} P(x''|x') f_1(x'y', x'')$ and $z_2(xy, x'y') = c(xy) + \gamma \sum_{x''} P(x''|x') f_2(x'y', x'')$, so that we can write

$$\|\mathcal{T}^\pi f_1 - \mathcal{T}^\pi f_2\|_\infty = \max_{xy, x'} \left| g_{q(\cdot|xy)}(z_1(xy, x'\cdot)) - g_{q(\cdot|xy)}(z_2(xy, x'\cdot)) \right|$$

$$\leq \max_{xy, x'} \|z_1(xy, x'\cdot) - z_2(xy, x'\cdot)\|_\infty = \|z_1 - z_2\|_\infty \leq \gamma \|f_1 - f_2\|_\infty,$$

where the last step follows from the straightforward calculation

$$\|z_1 - z_2\|_\infty = \sup_{xy, x'y'} \sum_{x''} P(x''|x') |f_1(x'y', x'') - f_2(x'y', x'')| \leq \|f_1 - f_2\|_\infty.$$

This concludes the proof.

$\square$

# F   Additional experimental results

In this appendix we present the results of additional experiments not included in the main text.

## F.1   Impact of the regularization parameter $\eta$

Besides the parameter $m$ that we have already studied experimentally in Section 5, the only tuning parameter of SVI and SPI is the regularization parameter $\eta$, which takes the role of a learning rate. In this experiment, we study two random walks run on two separate 4-room environments [Sutton et al., 1999], with two separate reward functions $r_{\mathcal{X}}$ and $r_{\mathcal{Y}}$ that together define the ground cost function $c(x, y) = |r_{\mathcal{X}}(x) - r_{\mathcal{Y}}(y)|$ for each state pair. The results of this study for $K = 2 \cdot 10^4$ iterations are shown in Figure 3. The error is computed as the difference between the distance estimate produced by the algorithm and a near-optimal distance obtained by running Algorithm 1 for a very small value of $\eta$ and a large number of iterations. We observe that higher values of $\eta$ lead to faster error reduction in the initial steps, but eventually prevent convergence to the true solution. In contrast, choosing smaller learning rates enables convergence to better solutions, at the cost of making slower progress initially. An intuitive explanation for this is that for larger values of $\eta$, the iterates converge rapidly to the broad proximity of an optimal solution, but then reach a cycle where they continue to perform large updates to the transition coupling which results in large constraint violations, which necessitate large updates, and so on. This is formally verified by the fact that our bound on the constraint violation terms in Theorem 4 are increasing for large values of $\eta$. Notably, employing a time-dependent learning-rate schedule (inspired by Ballu and Berthet [2023]) with $\eta_k \sim 1/\sqrt{k}$ leads to the best performance. This strategy leverages faster convergence initially and, for sufficiently large number of iterations, also achieves near-optimal solutions.

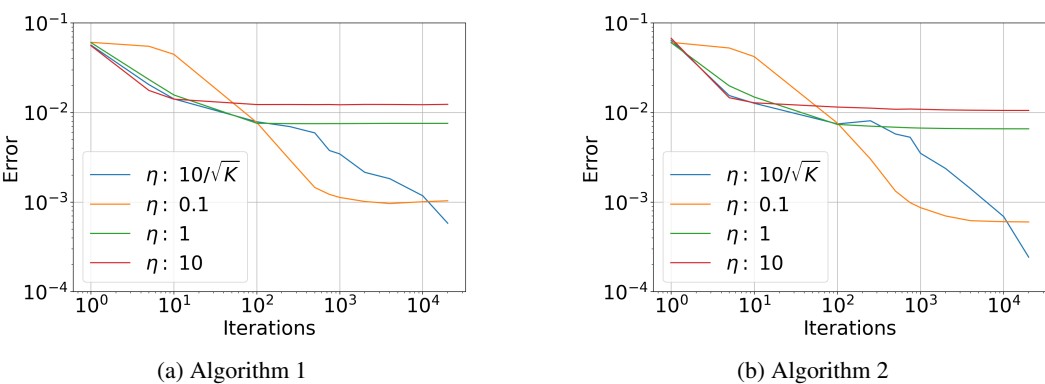

(a) Algorithm 1                                    (b) Algorithm 2

Figure 3: Error of estimated transport cost as a function $k$, for various choices of $\eta$.

## F.2   Comparison with alternative methods

We now turn to studying the computational complexity of our algorithms, and compare them empirically to some existing methods that have been proposed for computing optimal transport distances and bisimulation metrics between Markov chains. Specifically, we will focus here on two previous methods: the method of [O'Connor et al., 2022] that we refer to as "EntripicOTC" and Algorithm 2 of [Brugère et al., 2024] that we call "dWL". We adapt both of these methods with some minor changes to our setting. First, we remove one scaling factor from the transport cost in the definition of the distance defined by Brugère et al. [2024] so that it matches ours. Second, the algorithm of O'Connor et al. [2022] is originally defined for the infinite-horizon average-cost case, and thus we made appropriate changes to adapt it to the discounted case by replacing their approximate policy evaluation step by $T$ applications of the discounted Bellman evaluation operator. As pointed out in Appendix A.2, the resulting methods are closely related, and can be regarded as approximate dynamic programming methods for solving the MDP formulation of our optimal transport problem presented in Appendix B. We also recall that the algorithm proposed by Kemertas and Jepson [2022a] for the purpose of computing bisimulation metrics also falls into the same class of approximate dynamic programming methods, and nearly matches the method of [Brugère et al., 2024]. The comparison

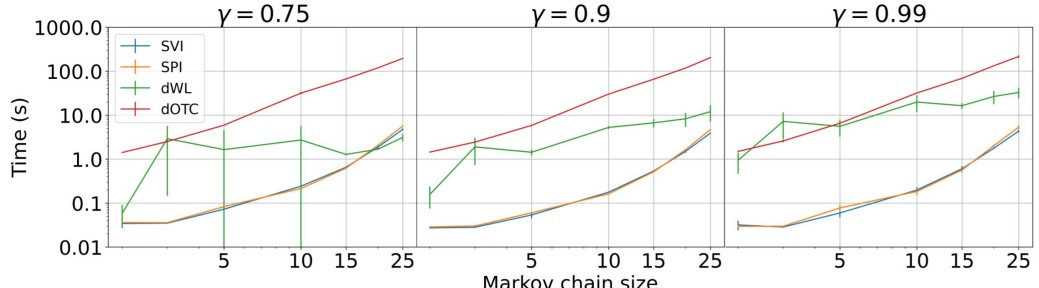

Figure 4: Comparison of the computational time of the different methods proposed to obtain a near-optimal solution for different values of $\gamma$. For each Markov chain size, the results obtained in 5 randomly generated instances are compared, showing the standard deviation in the plot. Data is displayed on a log-log scale.

below is based on the original Python implementation[3] of Brugère et al. [2024] and our own Python adaptation of the MATLAB code[4] of O'Connor et al. [2022].

One difficulty that we had to face in these experiments is having to tune various hyperparameters of each method (such as number of iterations and regularization parameter), which can each influence the quality of the solution and the computation time. For a fair comparison between the methods, we have adopted the following procedure to obtain our results. First, we estimate a ground truth obtained by running one of the algorithms for a very high number of iterations and a very low level of regularization, and then use this ground truth as a comparator to adjust the hyperparameters of the algorithms so that they are close to this value in as little wall-clock time as possible. While all the algorithms perform similar operations, their total runtimes turn out to be rather different and heavily dependent on problem parameters such as the discount factor $\gamma$. The comparison between the resulting runtimes of each method is shown on Figure 4 as a function of the size of the Markov chains, and for a three different choices of the discount factor, for a set of randomly generated MDPs (following the setup described in Section 7.1 of O'Connor et al., 2022).

First, we observe that EntropicOTC is a policy-iteration-like method, and as such it needs fewer iterations than the rest of the algorithms we tested to converge to the true cost, but each iteration requires running an expensive policy evaluation subroutine until convergence. This computational cost eventually adds up in a way that this algorithm has always ended up being the slowest among all that we have tested, although its performance has proved notably robust to changes in the discount factor $\gamma$. Second, we note that the updates of dWL are much cheaper to compute, especially for large regularization parameters. However, the errors of this value-iteration-like method compound much more rapidly than in the case of EntropicOTC, which makes it especially hard to tune the hyperparameters of this method. This problem is especially pronounced when the discount factor is large, which is the most interesting regime as it leads to distances with much stronger discrimination power.

The plots shown on Figure 4 indicate that our methods find optimal couplings consistently more efficiently than the other methods in the regime we studied, leading to up to 10 times faster runtimes. For the case of sufficiently small values of $\gamma$, the algorithm of Brugère et al. [2024] sometimes performs competitively, but the hyperparameters leading to good performance are much harder to find than in the case of our methods. In our experience, the massive speedup achieved by our methods can be largely ascribed to maintaining the transition couplings $\pi_k$ between iterations, as opposed to computing these afresh by running Sinkhorn's algorithm from scratch for each update as done by all other competing methods. Adjusting these other algorithms by maintaining the couplings in memory and using them to warm-start the subsequent updates makes them competitive with our methods, and in fact doing so makes them quite similar to SVI and SPI. Our algorithms use such warm-starts as a primary design choice as opposed to an obscure implementation detail, which is ultimately responsible for the computational efficiency of all these dynamic-programming methods.

---

[3]`https://github.com/yusulab/ot_markov_distances`
[4]`https://github.com/oconnor-kevin/OTC`

The computational time per iteration of each of these methods grows roughly at a rate of $n^4$, with $n$ being the number of states of the Markov chains. In the case of our methods, this is easily explained by noting that applying the Bellman–Sinkhorn operators and updating the transition couplings requires $|\mathcal{X}|^2|\mathcal{Y}|^2$ operations in total. This matches the runtime necessary for running the regularized policy improvement subroutines employed by O'Connor et al. [2022] and Brugère et al. [2024], which consists of running an instance of Sinkhorn's algorithm in each pair of states. The computational cost of all these methods can be improved by leveraging the sparsity of transition kernels: in particular, if at most $S$ states are reachable with positive probability in both of the chains, the complexity of the updates can be trivially improved to $|\mathcal{X}||\mathcal{Y}|S^2$. We did not pursue this direction in our experiments as our goal was to compare the basic versions of each studied method, and we believe that our conclusions would not be altered if we were to implement this improvement for all methods.

### F.3 Optimal transport distances as similarity metrics

We finally provide a range of experiments that illustrate how the optimal-transport distances we studied in this paper can be used to capture relationships and symmetries in groups of varying Markov chains. To this end, we have generated 35 different "4-room" instances and computed their pairwise distances. Each instance differs in its initial state and position of the obstacles (amounting to changes in the transition kernel), while maintaining a fixed reward function, with one reward located in each room except for the upper left room. Crossing a door between each room results in a negative reward. For each instance, we have studied the Markov chain induced by the corresponding optimal policy (which amounts to taking the shortest path toward the closest positive reward, modulo the additional randomness inherent to the transitions). Figure 5 presents a 3D visualization (where the $z$-dimension is represented by a color gradient) generated using Multidimensional Scaling (MDS) [Kruskal, 1964] based on the computed pairwise distances. One can observe a clear clustered structure, where instances with similar behaviors are grouped closely together. As the figure highlights, the resulting metrics capture the intuitive similarities and symmetries between each process, which indicates the potential usefulness of optimal transport distances and bisimulation metrics for comparing Markov chains under minimal structural assumptions made on the state spaces and the transition functions.

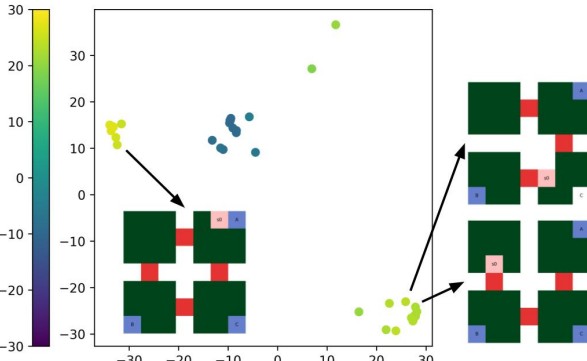

Figure 5: Result of applying MDS to the pairwise distances between the set of 4-room instances studied. On the plot, the first two coordinates of the MDS embedding are used as the spatial coordinates, and the third coordinate is encoded via the color bar provided on the left-hand side of the axes. It can be observed how the elements in the same cluster present common features that differentiate them from those in another cluster. In the examples shown in the figure we can see how the instances in which the closest reward involves crossing a door are concentrated in one cluster, while the instances in which the reward and the initial state are located in the same room belong to a different cluster. The remaining clusters correspond to having to cross two doors for a reward (set of green points on the top), or having no reward that is accessible from the initial state (set of blue points in the middle, with large negative $z$-coordinates).

