# OpenReview forum: "Bisimulation Metrics are Optimal Transport Distances, and Can be Computed Efficiently"
_NeurIPS.cc/2024/Conference — NeurIPS 2024 poster_

### Official Review · Reviewer_NKJ1 · 2024-07-09

**Soundness:** 3
**Presentation:** 3
**Contribution:** 3
**Rating:** 6
**Confidence:** 2

**Summary:**

The goal of this work is to study optimal transport (OT) distances between pairs of finite Markov chains, providing a novel relation between OT distances and probabilistic bisimulation metrics. The proposed linear program builds on ideas from optimal control in Markov decision processes, and the designed algorithm for solving the proposed linear program combines Sinkhorn's algorithm with an entropy-regularized version of the classic Value Iteration algorithm. Convergence guarantees and computational complexity analysis of the method are provided in the main text and in the appendix.

**Strengths:**

The paper is well-organized, the topic of study is very interesting, and the contribution is novel. Previous works on OT and bisimulation metrics are well-referenced and up-to-date.

**Weaknesses:**

The main theoretical contributions are presented in Section 3, where the authors introduce the definition of occupancy couplings $\mu$. These are distributions over $\mathcal{XY}\times \mathcal{XY}$. It is not entirely clear why they need to duplicate the variables $(x,y)$ and consider quadrupes $(xy,x'y')$.

**Questions:**

- In the abstract, the authors say "In this work, we develop an alternative perspective by considering couplings between a “flattened” version of the joint distributions that we call discounted occupancy couplings [...]" Could you expand on what do you mean by "flatten"?

- In the introduction, the authors say that "The possibility that the objects in question may be random further complicates the picture, and in such cases it becomes more natural to measure distances between the underlying joint probability distributions." Could you explain/expand?

- Section 2.1:

Does $\gamma$ depend on $t$?

Is $\gamma^t\in (0,1)$?

- Line 99, if Markov chains are considered, isn't it that $M_{\mathcal{X}}$ $(x_n | \bar{x}_{n-1})$ $=$ $M_{\mathcal{X}}$ $(x_n|x_{n-1})$ ?

- Line 148 and equation (4): Which is the dependence of $d_\gamma$ on the labeling function $r$?

Line 149: Which is the dependence of the set $\mathcal{F}_\gamma$ on $\gamma$?

It is clear from eq. (5) that $U^*$ does explicitly depend on $r$ and $\gamma$. Could you expand on the equality $U^*=d_\gamma$?

- Explain the pevious to last equality in (6). How do you make the variables $x',y'$ to apprear?

- Explain the last equality in (6). Why do you have $\langle \mu^\pi,c\rangle$ if c depends on $(x,y)$ and $\mu^\pi$ depends on $x,y,x',y'$?

Comments:

- The term "discounted occupancy couplings" mentioned in the abtract is not excatly used in the main text, the authors mainly use just "occupancy couplings".

- $n$ is used for the size $|\mathcal{Y}|$ (line 87) and as an index.

Isn't it $|\mathcal{X}|=m=|\mathcal{Y}|$? In the experiments, why the choice of $n$ is not analyzed too?

Typos:

- Abstract: line 17, "method that we call", instead of "method we call".

- Notations: line 79, "the corresponding finite subsequences", instead of "the corresponding subsequences"

- Line 39 (Introduction): "important" is repeated twice.

- Notations: Add the definitions of $\mathcal{X}^\infty$, $\mathcal{X}^n$, $\Delta_{\mathcal{X}}$.

---

> ### Author Rebuttal · Authors · 2024-08-05
>
> Thank you for your positive evaluation of our work and your very detailed reading! We will address your main points below, and will take the remaining minor comments into account when working on the final version of the paper.
>
> Q1: It is not entirely clear why they need to duplicate the variables $(xy)$ and consider quadruples $(xy,x'y')$.
>
> Indeed, defining occupancies as a function of $xy$ is perfectly sufficient for rewriting the optimization objective as a linear function. Duplication of the variables is necessary however for defining the constraints of Eq. (7-9): indeed, without the second set of variables is necessary for stating both the flow constraints and the transition coherence constraints. Consequently, without these additional variables, it would not be possible to characterize the set of occupancy measures via a set of linear constraints.
>
> Q2: Could you expand on what do you mean by "flatten"?
>
> We mean that occupancy couplings are low-dimensional projections ("flat" representations) of the joint distribution of the infinite sequence of state pairs. Our main result is showing that this representation is sufficient as long as one is concerned with calculating OT distances. We will clarify in the final version.
>
> Q3: ... Could you explain/expand?
>
> This sentence was probably not as nicely phrased as it should have been. We will refine it for the final version.
>
> Q4: Does gamma depend on t?
>
> No, it doesn't (as is normally the case in the theory of discounted MDPs).
>
> Q5: Is gamma^t \in (0,1)?
>
> Yes; we will state this more prominently.
>
> Q6: Line 99, if Markov chains are considered, isn't it that $(x_n | \bar{x}{n-1})=M{\mathcal{X}}(x_n|x_{n-1})$ ?
>
> True; this line merely served to lay down notation for general stochastic processes. We will clarify.
>
> Q7: Line 148 and equation (4): Which is the dependence of $d_\gamma$ on the labeling function $r$?
>
> The labeling function impacts the choice of the function class F. Thanks for pointing this out!
>
> Q8: Line 149: Which is the dependence of the set  $\mathcal{F}_\gamma$  on $\gamma$?
>
> Again, $\gamma$ appears in the concrete definition of the function class; we omitted this detail to maintain readability of the main text, but you are perfectly right that we should have at least hinted at this here. We will clarify.
>
> Q9: It is clear from eq. (5)...
>
> See our response to the previous two questions --- we will make this more clear in the final version.
>
> Q10: Explain the pevious to last equality in (6). How do you make the variables $(x'y')$ to apprear?
>
> We use the simple fact that $c(xy)$ does not depend on $x'y'$, and as such summing out $x'y'$ in the definition of $\mu(xy,x'y')$ results in the same discounted sum of indicators as what appears in the last line of Eq.6. We will add a remark.
>
> Q11: The term "discounted occupancy couplings" mentioned in the abtract is not excatly used in the main text, the authors mainly use just "occupancy couplings".
>
> Thank you for pointing this out, we will smooth out the terminology!
>
> Q12: $m$ is used for the size (line 87) and as an index.
>
> Thanks for calling this to our attention! This is an unfortunate clash of notation that we didn't notice before and will fix it. In the experiments, we have used $m$ to denote the number of applications of the Bellman--Sinkhorn operators (see Alg 1).

---

> > ### Comment · Reviewer_NKJ1 · 2024-08-12
> >
> > The authors have gone over all my comments and their answers to my questions are very precise.
> > The only exception is Q3 (which is a minor issue): Although the authors didn't expand/explain the phrase included in the Introduction, they say that theyr will refine it in the last version (and I'd appreciate it).
> > I maintain my acceptance rate and I thank the authors very much.

---

### Official Review · Reviewer_VaVz · 2024-07-12

**Soundness:** 3
**Presentation:** 3
**Contribution:** 3
**Rating:** 7
**Confidence:** 3

**Summary:**

This submission

Namely, the authors define a notion of optimal transport distance between Markov chains on state spaces with a ground metric. This notion of distance differs from standard optimal transport, as the set of couplings is restricted to the set of so-called bicausal couplings. Using the results of Moulos [2021], it is shown that this Markovian optimal transport can be characterized in terms of the solution of the Bellman optimality equations for a Markov decision process.

Using this equivalence, it is demonstrated that, in certain cases, bisimulation metrics can be though of, equivalently, as Markovian optimal transport with a specific cost function by exploiting their connection to solutions of certain fixed-point equations.

Next, it is shown that the optimal value for the derived Markovian optimal transport problem can be computed by solving a finite-dimensional linear program, where the constraint set consists of the intersection of three sets. Rather than directly solving this linear program, the authors propose to regularize the problem using a conditional entropy (by analogy with entropic regularization of optimal transport distances which can then be solved efficiently using Sinkhorn iterations). To solve the entropy regularized problem, the Sinkhorn Value Iteration algorithm is proposed and the number of steps required to obtain a desired accuracy in estimating the Markovian optimal transport distance is provided. The authors also propose an analyze an alternative algorithm which they dub Sinkhorn Policy Iteration.

The paper concludes with some numerical experiments to illustrate the performance of the proposed algorithms.

**Strengths:**

In my opinion, the submission is well-written and its contributions relative to the broader literature are clearly identified.

The main contributions of this work, identifying bisimulation metrics as a type of optimal transport problem and providing some new algorithms for estimating bisimulation metrics by using this connection is of interest and is, to my knowledge, novel.

**Weaknesses:**

1. It appears that the connection between the Markovian optimal transport problem and the finite-dimensional linear program provided in Theorem 1 only enables the computation of the optimal value for optimal transport problem, but does not allow the recovery of the optimal bicausal coupling. If this is the case, this should be further clarified in the text.

2. While the analysis for the Sinkhorn Value Iteration is a nice addition, the guarantees are a bit confusing. Notably, if we wish to attain a precision of $\epsilon$ in estimating the Markovian optimal transport one requires $K=O(1/\epsilon^2)$ iterations and setting the regularization parameter to $\eta =C/\sqrt K=O(\epsilon)$. However, if epsilon is sufficiently small, the objective in the update (10) will be dominated by the entropy which is minimized by $\mu=\mu_k$. Naively I would assume that $1/\eta$ should be small so that minimization of the linear term dominates.

**Questions:**

Apart from the points mentioned above, it would be helpful for the authors to state the complexity of the Sinkhorn Value Iteration algorithm in the main text (rather than only in the appendix).

**Limitations:**

The authors address the limitations of their work in the discussion.

---

> ### Author Rebuttal · Authors · 2024-08-05
>
> Thank you for your positive comments and your critical reading of our work! Regarding your questions:
>
> 1. Note that the argument $\mu^*$ achieving the infimum exists, and an optimal transition coupling $\pi^*$ can be decoded from it. Concretely, given the joint distriibution $\mu^*$ over $\mathcal{X}\mathcal{Y}\times\mathcal{X}\mathcal{Y}$, the corresponding transition coupling can be extracted as $\pi^*(x'y'|xy) \propto \mu^*(xy,x'y')$ (see also line 776 in the appendix for the general formula relating occupancy couplings to transition couplings). We will clarify this in the final version. In any case, we note that our algorithms (SVI and SPI) do return transition couplings besides the transport distances.
>
> 2. This observation is entirely correct: the individual updates do get smaller and smaller as $epsilon$ goes to zero. Note however that, at the same time, the number of iterations $K$ goes to infinity at a rate of $1/\epsilon^2$, and as such the algorithm will make more and more updates that get smaller and smaller as the desired precision approaches zero. This is normal for algorithms that use a fixed learning rate. Following Ballu & Berthet (2023), it would be possible to extend our analysis to time-varying learning rates that start out large and then decay to zero over time, and we have observed in our experiments that such learning rates are indeed easier to tune and work with. We will add a more prominent comment about this in the final version.
>
> Re complexity of SVI: This is a great point, we will update the paper accordingly for the final version!

---

> > ### Comment · Reviewer_VaVz · 2024-08-09
> >
> > Thank you for clarifying these points, I am content with the response.

---

### Official Review · Reviewer_aiTP · 2024-07-12

**Soundness:** 3
**Presentation:** 3
**Contribution:** 3
**Rating:** 6
**Confidence:** 2

**Summary:**

This work integrates optimal transport with Markov chains by proposing an alternative joint distribution between Markov processes, namely "discounted occupancy couplings". They show that optimal transport distances can be computed as a a linear program (LP) in reduced space. This improves the computational efficiency of OT between markov decision process (MDP) over the previous methods. The previous methods, as they reviewed, often requires complex dynamic programming algorithms. They showed that the new formulation can be extended to the well-known entropy regularization problem, employing Sinkhorn-like iterations to solve, making the computation scalable with large problem. This paper provides both theoretical and experimental supports for the new formulation. In the end, they discuss the potential applications on RL models, limitations and challenges.

**Strengths:**

The main contribution of this work is demonstration of optimal transport shares the same formulation as probabilistic bisimulation metrics, which is popular in the practice of reinforcement learning (RL). By establishing that solving for optimal transport is equivalent to computing bisimulation metrics, the authors creates a novel link between RL and optimal transport theory.
- The formulation of OT on MDP is novel. This work provides an new prospective of Bisimulation metric between MDPs.
- Theoretical analysis looks good.
- Two computational solutions (Sinkhorn-like iterations) were proposed and tested in experiments sections.

**Weaknesses:**

The theoretical guarantees are based on the assumption of perfect projection steps $m=\infty$, which is not practical. As stated in line 290, such exact computation is unnecessary in practice. This claim needs a further theoretical support.

**Questions:**

- What is the best practice of bisimulation metric computation? Could your provide the comparison with your methods?
- What does the entropy means in the sense of the coupling between MDPs? Some explanations on this will be appreciated
- How does this metric works with data contains noise as we know OT is sensitive to outliers?

---

> ### Author Rebuttal · Authors · 2024-08-05
>
> Thank you for your positive evaluation of our work, as well as your insightful remarks! We respond to your questions below.
>
> Re weakness: We agree that the analysis working only for the case $m=\infty$ is the biggest limitation of our results. By analogy with modified policy iteration (that bridges the cases $m=1$ and $m=\infty$ for standard dynamic programming), we expect that our results should generalize to finite values of $m$, but at the moment we do not have a proof. We nevertheless find the empirical results shown on Figure 1 to be encouraging. In any case, we hope that we were clear enough in stating this limitation in the paper, and hope that you agree that our results are still worthy of publication despite this limitation.
>
> Q1) What is the best practice of bisimulation metric computation? Could your provide the comparison with your methods?
>
> The best method we are aware of for computing bisimulation metrics is due to Kemertas and Jepson (2022), which essentially aims to approximately solve Eq. (5) by using Sinkhorn's method to approximate the infimum on the right-hand side. Their method is essentially the same as the algorithm of Brugere et al. (2024) that we have discussed in some detail in the appendix. All similar methods are significantly slower in practice than our method. We will add further discussion about these matters to the final version of the paper.
>
> Q2) What does the entropy means in the sense of the coupling between MDPs? Some explanations on this will be appreciated.
>
> The entropy we use is the Bregman divergence induced by the conditional entropy of $X',Y'$ given $X,Y$ if the tuple $XY,X'Y'$ is drawn from the occupancy measure. The conditional entropy (sometimes called the "causal entropy") plays a key role in the theory of entropy-regularized Markov decision processes, and can be shown to be the "correct" notion of entropy-regularization in that it induces the well-known "soft Bellman equations" as its dual (Neu, Jonsson, Gomez, 2017). We will add a few explanatory lines to the final version of the paper.
>
> Q3) How does this metric works with data contains noise as we know OT is sensitive to outliers?
>
> This is a great question! So far we have not thought about this, as we have focused on computing distances between perfectly known Markov processes in this work. It remains to be seen how to compute distances between Markov chains based purely on data (that may include noise or outliers). In any case, we are sure that developing algorithms for this more realistic case will require the foundations laid down in the present paper.

---

> > ### Comment · Reviewer_aiTP · 2024-08-13
> >
> > I appreciate your response and clarification. I think your work is worth for publication.

---

### Decision · Program_Chairs · 2024-09-25

**Decision:**

Accept (poster)

**Comment:**

Reviewers are unanimous that this is an interesting submission and that it should be accepted.